# TREX reveals proteins that bind to specific RNA regions in living cells

Martin Dodel [1,3], Giulia Guiducci [1,3], Maria Dermit[1], Sneha Krishnamurthy [1], Emilie L. Alard [1], Federica Capraro [1,2], Zeinab Rekad [1], Lovorka Stojic [1] ✉ & Faraz K. Mardakheh [1] ✉

Different regions of RNA molecules can often engage in specific interactions with distinct RNA-binding proteins (RBPs), giving rise to diverse modalities of RNA regulation and function. However, there are currently no methods for unbiased identification of RBPs that interact with specific RNA regions in living cells and under endogenous settings. Here we introduce TREX (targeted RNase H-mediated extraction of crosslinked RBPs)—a highly sensitive approach for identifying proteins that directly bind to specific RNA regions in living cells. We demonstrate that TREX outperforms existing methods in identifying known interactors of U1 snRNA, and reveals endogenous region-specific interactors of NORAD long noncoding RNA. Using TREX, we generated a comprehensive region-by-region interactome for 45S rRNA, uncovering both established and previously unknown interactions that regulate ribosome biogenesis. With its applicability to different cell types, TREX is an RNA-centric tool for unbiased positional mapping of endogenous RNA–protein interactions in living cells.

The fate and function of RNA molecules in living organisms are defined by their interactions with RBPs. RNA–RBP interactions, therefore, play a fundamental role in regulating all aspects of cell behavior. Several methods have been developed for deciphering RNA–RBP interactions in living cells. These can be broadly divided into protein-centric approaches, which reveal the RNAs that are bound by a specific RBP, or RNA-centric approaches, which reveal the RBPs that bind to a specific RNA[1]. Crosslinking and immunoprecipitation (CLIP)-based methods are among the most powerful and widely used protein-centric approaches, which leverage the sensitivity of next-generation sequencing to not only profile the compendium of RNAs that directly interact with a given RBP, but also map their precise binding sites[2–5]. Transcriptome-wide binding sites of hundreds of RBPs have been profiled using CLIP-based approaches[6]. A more recent protein-centric approach for RNA–RBP profiling involves fusion of RBPs with RNA-editing enzymes, followed by transcriptome-wide identification of modified RNAs via next-generation sequencing[7,8]. By eliminating the need for immunoprecipitation, these methods have pushed the limits of protein-centric RNA–RBP profiling down to the single-cell level[8].

In contrast, RNA-centric profiling of RNA–RBP interactions has remained more challenging. Most common current methods use crosslinking, followed by RNA affinity capture and mass spectrometry (MS) analysis[1]. For example, biotinylated antisense oligonucleotides coupled with streptavidin-conjugated beads have been used to profile the interactomes of U1, XIST, NORAD, 18S and RMRP RNAs, among others[9–12]. Low efficiency has been a main caveat of these methods, often requiring large numbers of cells for successful identification of interacting RBPs[13]. An alternative strategy involves the use of proximity-based labeling enzymes, such as biotin ligase BirA or ascorbate peroxidase, targeted to a specific RNA of interest[14–18]. Subsequently, the labeled proteins are captured and identified by MS. A notable limitation of this approach is the inclusion of indirectly associated proteins that are in close physical proximity to the targeted RNA. Most importantly, neither method can reveal which RNA regions are responsible for mediating different RNA–RBP interactions.

[1]Centre for Cancer Cell and Molecular Biology, Barts Cancer Institute, Queen Mary University of London, London, UK. [2]Randall Centre for Cell and Molecular Biophysics, King's College London, London, UK. [3]These authors contributed equally: Martin Dodel, Giulia Guiducci. ✉e-mail: l.stojic@qmul.ac.uk; f.mardakheh@qmul.ac.uk

To address these shortcomings, we developed TREX, which provides a highly efficient method to extract RBPs that are bound to a specific RNA sequence, followed by their identification by quantitative MS. Unlike previous RNA-centric methods, TREX can map endogenous RNA–protein interactions in a region-specific manner. We benchmarked TREX against the full length of U1 small nuclear RNA (snRNA), the well-characterized RNA component of the U1 spliceosome complex[19], demonstrating its superiority over existing antisense oligonucleotide-based pulldown methods for revealing known RNA–protein interactions. Next, we applied TREX to the most conserved segment of NORAD long noncoding RNA (lncRNA), highlighting its versatility in probing region-specific interactions in endogenous settings. Finally, we used TREX to generate a detailed region-by-region interactome map for the 45S ribosomal RNA (rRNA). This mapping unveiled many established as well as several previously unknown interactions, providing insights into the intricate regulation of ribosome biogenesis and composition. Our findings establish TREX as a versatile RNA-centric approach for revealing region-specific RNA–RBP interactions.

## Results

### Integrating organic phase separation with RNase H targeting

Several recent studies have reported that ultraviolet (UV)-C crosslinking in combination with acid guanidinium thiocyanate-phenol-chloroform extraction can be used to isolate and purify RBPs that directly bind RNA[20–22]. UV-C covalently crosslinks RNA molecules to their interacting RBPs in living cells. Subsequent cell lysis in acidic guanidinium thiocyanate-phenol (commonly known as TRIZOL) results in disruption of membranes and denaturation of macromolecules. The lysate is then subjected to organic phase separation by adding chloroform, leading to the partitioning of free RNA molecules to the upper aqueous phase and free proteins to the lower organic phase. RNA–RBP adducts, however, cannot partition into either phase and form an insoluble interface that can be isolated and subjected to proteomics analysis for global profiling of the RNA-bound proteome[20,21].

In TREX, we utilize a similar approach to isolate RNA–RBP complexes, but the interface is resolubilized before denaturation and annealing of non-overlapping tiling antisense DNA oligonucleotides that are fully complementary to an RNA sequence of interest. The average length of these oligonucleotides is 60 bases. This is comparable with the probe lengths used in ribodepletion for RNA sequencing library preparations, which are optimal for effective removal of specific RNA species with minimal bias and off-target effects[23,24]. The targeted RNA sequence, now hybridized to DNA, is specifically degraded by RNase H, resulting in the release of any crosslinked RBPs. A second organic phase separation is then performed to partition the released RBPs into the organic phase, allowing their extraction from the remaining RNA-bound RBPs, and subsequent analysis by quantitative MS (Fig. 1a). We confirmed that intact total RNA from the interface can be effectively solubilized for downstream processing (Extended Data Fig. 1a,b). Moreover, upon RNase H addition, tiling antisense oligonucleotides against several different types of RNA were confirmed to specifically deplete their targets in a dose-dependent manner (Extended Data Fig. 1c–f). These findings indicate that the crucial steps of TREX, namely interface solubilization and RNase H-mediated target degradation, operate efficiently to recover and selectively deplete protein-bound RNA sequences of interest.

### TREX outperforms existing methods in defining U1 interactors

Next, we benchmarked TREX by applying it to analyze the direct binding partners of U1 snRNA (Fig. 1b), the 164 base (b)-long RNA component of the U1 small nuclear ribonucleoprotein particle (snRNP), whose protein interactions are well characterized[19]. Human U1 snRNP contains ten proteins, including seven core spliceosome components

(SNRPB, SNRPD1, SNRPD2, SNRPD3, SNRPE, SNRPF and SNRPG), and three U1-specific factors (SNRPA, SNRPC and SNRNP70). Ten million HCT116 human colorectal carcinoma cells per biological replicate were used, and, as negative controls, equivalent preparations were performed but without the addition of RNase H. To assess the efficiency of the RNase H-mediated degradation, RNA was isolated from a portion of each reaction before the second organic phase separation (Fig. 1a). Quantitative PCR with reverse transcription (RT–qPCR) analysis confirmed the effective removal of U1 in the RNase H-treated samples (Extended Data Fig. 1g). The specificity of RNase H-mediated degradation was also evaluated through whole-transcriptome RNA sequencing, which showed various U1 isoforms as the main depleted transcripts, followed by SNORA74A and B, two small nucleolar RNA (snoRNA) isoforms that share over 40% sequence similarity with U1 (Extended Data Fig. 1h). Importantly, SNORA74A and B were present in amounts >20-fold lower than those of the U1 isoforms (Extended Data Fig. 1h), suggesting that most of the released proteins are likely to come from U1.

After validating the efficiency and specificity of U1 degradation, the remainder of each reaction was subjected to the second organic phase separation to isolate the released proteins, which were subsequently recovered, digested with trypsin and analyzed by quantitative MS (Fig. 1a). Principal component analysis (PCA) of the results indicated high reproducibility among the RNase H-treated biological replicates (Extended Data Fig. 1i). A total of 27 significant direct U1 interactors were identified, including 8 out of the 10 U1 snRNP components, along with SF3A1 and ADAR—two other known interactors of U1 snRNP[25,26] (Fig. 1c,d and Supplementary Dataset 1). Proteins specific to other snRNP complexes (for example, U2, U4, U5 and U6), however, were not identified as significant interactors, showcasing the specificity of TREX (Fig. 1e and Supplementary Dataset 1).

Since the U1 interactome has been investigated previously in a number of RNA pulldown-based studies, we were able to evaluate the performance of TREX against these[9,10,12]. Despite starting from a lower number of cells as input, TREX was able to recover more of the known U1 snRNP proteins than any other previous study (Fig. 1f), demonstrating its superiority for revealing true interactions in comparison with state-of-the-art RNA pulldown techniques.

### TREX identifies the interactors of the ND4 segment of NORAD

Encouraged by the TREX analysis of U1 snRNA, we set out to investigate the capacity of TREX to decipher region-specific interactions. To address this, we focused on NORAD, a 5.3 kilobase (kb)-long lncRNA that is highly conserved and expressed robustly in most human tissues and cell lines[27,28]. NORAD is upregulated in response to DNA damage, and its main reported function is to safeguard genome stability[11,28]. It localizes predominantly to the cytoplasm, but is also present in the nucleus[11,27–29]. The cytoplasmic translational regulators Pumilio 1 and 2 (PUM1 and PUM2) constitute the main known functional interactors of NORAD[27,28,30,31]. At the primary sequence level, NORAD consists of five repetitive segments of ~400 b long known as NORAD domains (ND1–ND5)[28], with the ND4 domain representing the most conserved segment (Fig. 2a). Although several studies have assessed the interactome of NORAD through different methodologies[11,27,28,31,32], identification of the direct interactors of the conserved ND4 segment under endogenous conditions has not been possible up to now.

We employed TREX to investigate the direct interactors of the ND4 segment of NORAD. First, we used RT–qPCR to validate the efficiency and specificity of the RNase H-mediated targeting of the ND4 region (Extended Data Fig. 2a,b). Subsequently, we conducted quantitative MS analysis from 100 million HCT116 cells per replicate experiment. PCA confirmed the reproducibility among biological replicates, with the exception of one RNase H-treated sample, which was removed from further analysis (Extended Data Fig. 2c). Overall, we identified 360 proteins as significant direct interactors of the ND4 region. This dataset

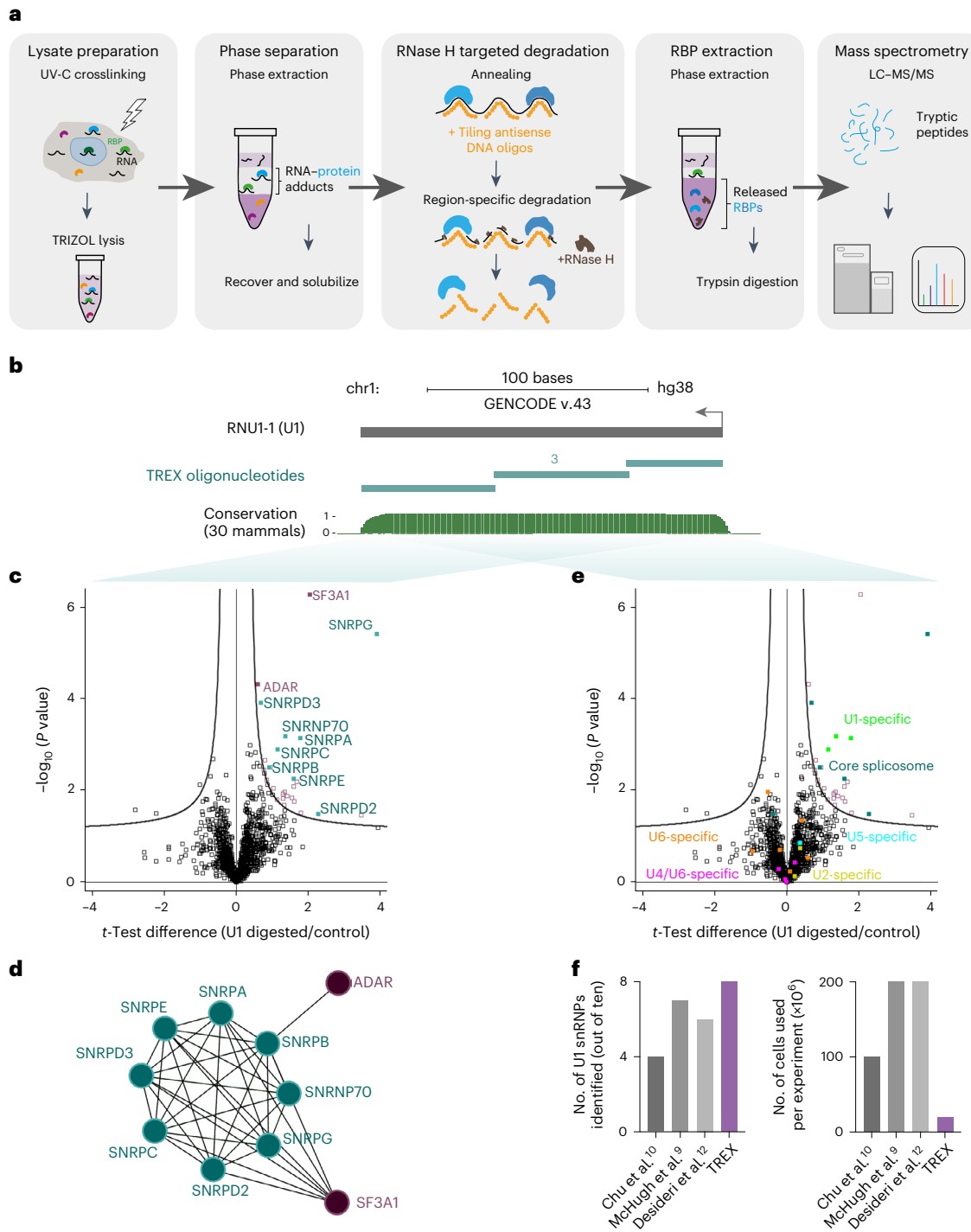

**Fig. 1 | TREX reveals proteins that bind to specific RNA sequences in living cells. a**, Experimental scheme of TREX. **b**, Schematic representation of U1 snRNA primary sequence (annotated as RNU1-1 in RefSeq) with the associated University of California Santa Cruz (UCSC) Genome Browser tracks for mammalian conservation (PhastCons). The tiling antisense DNA oligonucleotides used for depletion of U1 snRNA in TREX (three in total) are depicted on the tracks. Scale, 100 bases. chr, chromosome. **c**, Volcano plot of the two-sided two-sample $t$-test comparison of U1 digested versus undigested TREX samples ($n = 5$ biological replicates), showing significant enrichment of the U1 snRNP complex members, along with SF3A1 and ADAR in the digested samples. Curved lines mark the significance boundary (FDR = 0.05, S0 = 0.1). **d**, Interaction network analysis of

U1-bound proteins identified by TREX, using the STRING physical interactions database[42]. **e**, Volcano plot of the two-sided two-sample $t$-test comparison of U1 digested versus undigested TREX samples ($n = 5$ biological replicates), with marking the core spliceosome as well as each identified specific U1 snRNP complex protein members. Only U1-specific proteins, as well as the core spliceosome components, are found as significantly enriched in the U1 digested TREX samples. Curved lines mark the significance boundary (FDR = 0.05, S0 = 0.1). **f**, Comparison of the number of cells used per experiment, against the number of known U1 snRNP proteins identified, in TREX versus three previous U1 RNA affinity capture–MS studies[9,10,12].

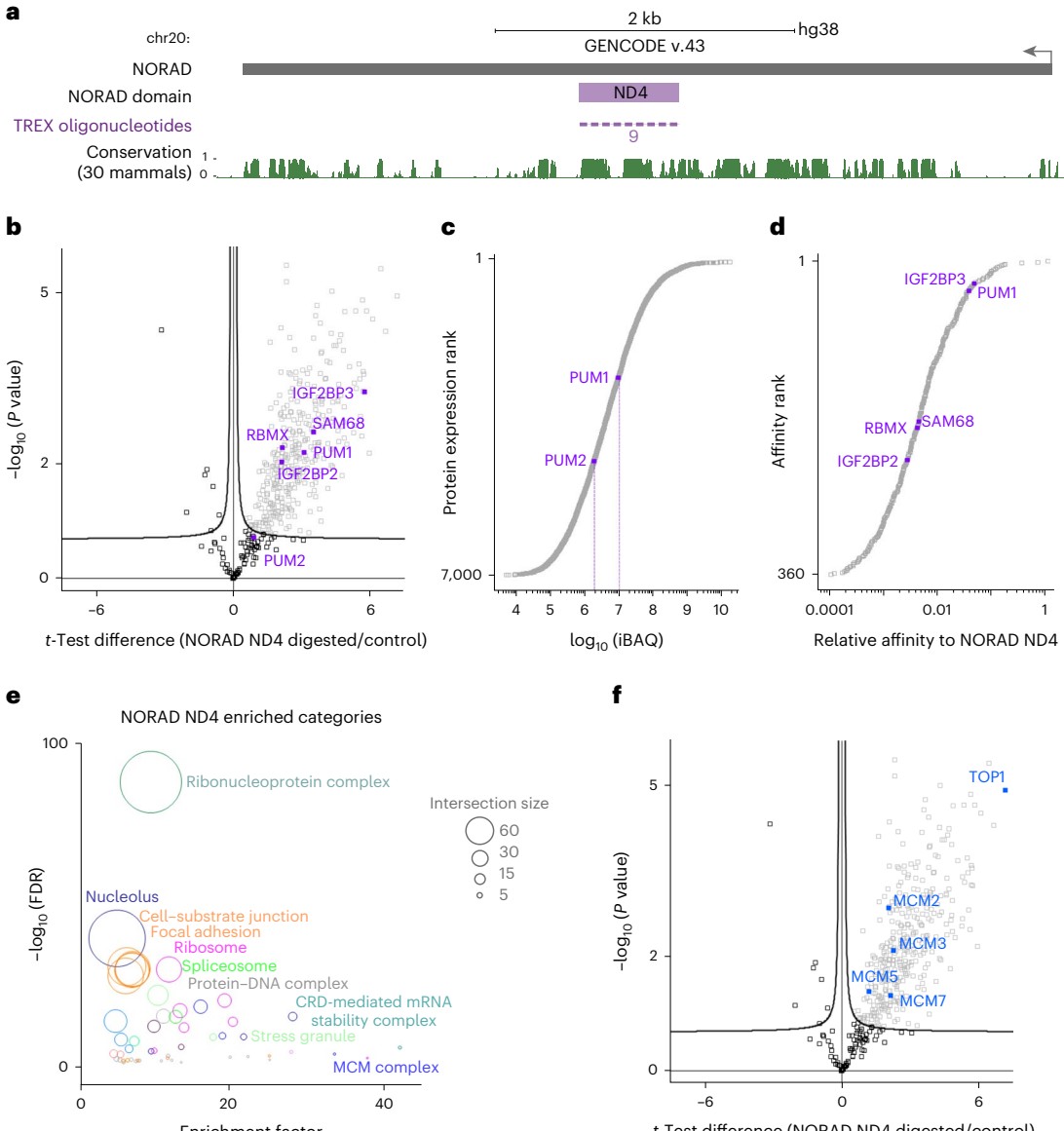

**Fig. 2 | TREX defines proteins that bind to the ND4 segment of NORAD lncRNA. a**, Schematic representation of NORAD lncRNA primary sequence and its ND4 segment, with the associated UCSC Genome Browser tracks for mammalian conservation (PhastCons). The tiling antisense DNA oligonucleotides used for depletion in TREX (nine in total) are depicted on the tracks. Scale, 2 kb. **b**, Volcano plot of the two-sided two-sample $t$-test comparison of NORAD ND4 digested versus undigested TREX samples ($n = 3$ biological replicates), showing the enrichment of several known RBP interactors of NORAD (purple) in the digested samples. Curved lines mark the significance boundary (FDR = 0.05, S0 = 0.1). While PUM1 is a highly significant interactor, PUM2 falls just below significant cut-off. **c**, The ranked plot of the iBAQ[34] absolute protein abundance measurements from the total proteome of HCT116,

revealing PUM1 to be expressed at amounts more than fivefold that of PUM2. **d**, The ranked plot of the iBAQ-based estimated relative affinities of the TREX-identified NORAD ND4 interactors, with several known RBP binding partners of NORAD marked on the graph (purple). **e**, Fisher's exact test analysis of known protein categories that are over-represented among the NORAD interacting proteins (Benjamini–Hochberg FDR < 0.05). Each circle represents an enriched category from the Gene Ontology Cellular Compartments (GOCC) database, with circle size representing the number of shared proteins. **f**, Volcano plot of the two-sided two-sample $t$-test comparison of NORAD ND4 digested versus undigested TREX samples ($n = 3$ biological replicates), with TOP1 and members of the MCM helicase complex highlighted. Curved lines mark the significance boundary (FDR = 0.05, S0 = 0.1).

included several previously reported NORAD interactors such as PUM1 and RBMX[11,27,28,33] (Fig. 2b and Supplementary Dataset 2). PUM2 did not reach significance in our analysis, possibly due to its lower expression in HCT116 cells, as estimated by intensity-based absolute quantitation (iBAQ)[34] (Fig. 2c). Interestingly, since the RNase H-mediated depletion of the ND4 region was almost complete in these experiments (Extended Data Fig. 2a), we could utilize iBAQ to estimate the relative affinity of the identified hits for this region, by calculating the protein amounts in the TREX eluates relative to the input total cell lysate. This analysis revealed PUM1 among the highest affinity interactors of the ND4 region

(Fig. 2d)—an observation in line with the proposed mechanism of NORAD to sequester Pumilio proteins[35].

Gene ontology analysis of the ND4 interactors revealed a diverse range of nuclear and cytoplasmic RNA- and DNA-associated proteins (Fig. 2e). These interactors seemed to be split evenly between cytoplasmic and nuclear protein families (Extended Data Fig. 2d), aligning with the reported localization NORAD to both compartments. We also compared the overlap between our data and the results of other studies that have assessed the NORAD interactome through methods such as in vitro RNA pulldown coupled with MS[27,28], hybridization-proximity

coupled with MS[32] and RNA antisense purification coupled with MS[11]. We observed a highly significant overlap, with ~18–44% of the identified interactors in these studies matching to our TREX results (Extended Data Fig. 2e).

Crucially, our analysis revealed topoisomerase I (TOP1) as one of the most enriched interactors of the ND4 region of NORAD (Fig. 2f and Supplementary Dataset 2). A recent study has proposed a role for NORAD in regulation of TOP1 function, but this has been proposed to occur indirectly via RBMX[11]. We employed CLIP coupled with RT–qPCR to orthogonally validate the direct binding of TOP1 to the ND4 region. Our CLIP–qPCR approach confirmed the interactions of PUM1 and RBMX, as well as validating the direct binding of TOP1 (Extended Data Fig. 2f–j). Additionally, TREX revealed several subunits of the minichromosome maintenance (MCM) replication helicase MCM2-7 that play a central role in DNA replication initiation and elongation as direct interactors of the ND4 region (Fig. 2e,f and Supplementary Dataset 2). Considering that TOP1 is known to bind to active replication origins and copurifies with the MCM complex[36], we speculate that NORAD is probably in direct contact with a TOP1–MCM complex. Collectively, these results demonstrate the effectiveness of TREX in identifying the functional interactors of a given RNA region under endogenous settings.

### TREX reveals the compendium of human 45S interactions

Next, we employed TREX to identify proteins that interact with both premature and fully processed rRNA molecules that derive from the *45S* gene in human cells. The 45S pre-rRNA is transcribed in the nucleolus as a transcript of ~13.4 kb long containing the 18S, 5.8S and 28S rRNA sequences, interspersed with four spacer sequences known as 5′ external transcribed spacer (5′-ETS), internal transcribed spacer 1 (ITS1), internal transcribed spacer 2 (ITS2) and 3′-ETS (Fig. 3a). Through a complex series of molecular events, 45S pre-rRNA undergoes extensive processing, resulting in the removal of spacer regions and the formation of mature ribosomal subunits containing the 18S, 5.8S and 28S rRNA molecules[37,38]. To identify the complete set of proteins interacting with the RNA products of the 45S locus, we employed TREX with tiling antisense DNA oligonucleotides spanning the entire length of 45S pre-rRNA (Fig. 3a). As before, the efficiency of RNase H-mediated depletion was validated by RT–qPCR (Extended Data Fig. 3a), while the specificity was assessed by whole-transcriptome RNA sequencing, revealing RPS26P58 pseudogene RNA as the only significant off-target hit (Extended Data Fig. 3b).

Subsequently, we conducted quantitative MS analysis of the 45S TREX samples, originating from 30 million HCT116 cells per replicate experiment. PCA revealed good reproducibility among the biological replicates (Extended Data Fig. 3c). We identified 160 proteins as significant direct interactors of 45S rRNA, 97 of which were already known to interact with rRNA/ribosomes (Fig. 3b and Supplementary Dataset 3). Among them were large ribosomal subunit proteins (RPLs), small ribosomal subunit proteins (RPSs), ribosome associated factors (RAFs) and known ribosome biogenesis factors (RBFs) (Fig. 3b,c). To further investigate the remaining 63 interactors, we compared our dataset with hits from three RNAi screens targeting functional regulators of human ribosome biogenesis, including two genome-wide screens and one nucleolar proteome-specific screen[39–41]. This revealed a further nine proteins that had a role in ribosome biogenesis (Fig. 3b,c). Furthermore, protein interaction network analysis using the STRING database[42] showed that 153 of the 160 identified proteins physically associate with each other (Fig. 3d). These interactions include most of the remaining proteins that lacked any known association to ribosomes, suggesting that they are probably previously unknown factors involved in rRNA-related processes. Together, these results demonstrate the ability of TREX to reveal the full repertoire of proteins that interact with 45S rRNA in living cells.

### TREX defines the interactors of 18S, 5.8S and 28S

We next set out to elucidate a region-by-region interactome of 45S rRNA. First, we focused on mapping the interactomes of the 18S (~1.8 kb), 5.8S (157 b) and 28S (~5 kb) rRNA regions by selecting tiling oligonucleotides from the 45S rRNA TREX that correspond to these segments (Fig. 4a). As before, the efficiency of each region-specific depletion was confirmed by RT–qPCR (Extended Data Fig. 4a–c). Subsequently, we conducted quantitative MS analysis of the TREX samples originating from ten million HCT116 cells per replicate experiment. Consistent with its role as the backbone of the 40S ribosomal subunit, we identified small ribosomal subunit proteins as the primary group of proteins that were directly bound to 18S rRNA (Fig. 4b,c and Supplementary Dataset 4). Several known small subunit RAFs, as well as specific RBFs involved in small subunit biogenesis[38], were also among these significant interactors (Fig. 4b). Interestingly, RPL24, a large ribosomal subunit protein, was also identified as a direct interactor of 18S rRNA. This finding aligns with the resolved structure of the human 80S ribosome[43], which shows extensive post-translocation contacts between the C-terminal portion of RPL24 and 18S rRNA (Extended Data Fig. 4d).

TREX analysis of 5.8S rRNA, also originating from ten million HCT116 cells per replicate, revealed a notably smaller set of interacting proteins (Fig. 4d and Supplementary Dataset 5). Among them, we identified DDX27—a DEAD box helicase known for its specific involvement in 5.8S processing[38]. RPL35 and RPL23A were the sole ribosomal proteins found to interact with 5.8S rRNA (Fig. 4d)—an observation that is consistent with the resolved structure of the human ribosome[43] (Extended Data Fig. 4e). Interestingly, our analysis also revealed UBR7—a nuclear E3 ubiquitin-protein ligase—as a strong interactor (Fig. 4d). We validated the direct binding of UBR7 to 5.8S rRNA by CLIP–qPCR (Extended Data Fig. 4f,g). Crucially, an interaction could also be detected with an ITS1 segment immediately adjacent to the 5.8S region (Extended Data Fig. 4g), suggesting that UBR7 is probably associated with unprocessed 5.8S rRNA. To examine the role of UBR7 in 5.8S biogenesis, we depleted it by RNAi, and assessed the impact on the steady-state levels of ITS1, ITS2 and 5.8S regions. UBR7 depletion resulted in significant accumulation of ITS1 and ITS2, and a concomitant decrease in the 5.8S levels (Extended Data Fig. 4h,i). These results present UBR7 as a previously unknown RBF that is involved in 5.8S rRNA processing.

In contrast to 5.8S rRNA, TREX analysis of 28S rRNA from the same number of HCT116 cells unveiled numerous interactors, primarily comprising large ribosomal subunit proteins, as well as RAFs that interact with the large ribosomal subunit[44,45], and various large subunit RBFs[38] (Fig. 4e,f and Supplementary Dataset 6). Our analysis also revealed some small ribosomal subunit proteins, most of which are positioned at the interface with the large ribosomal subunit (Extended Data Fig. 4j). Interestingly, our findings revealed evidence of heterogeneity in the ribosomes of HCT116 cells, generated by incorporation of distinct ribosomal protein isoforms. Specifically, RPL22 and RPL22L1 were both found to interact with 28S rRNA (Extended Data Fig. 4k), indicative of their successful incorporation into ribosomes. In contrast, although both RPL7 and RPL7L were expressed in HCT116 cells, only RPL7 was associated significantly with 28S rRNA (Extended Data Fig. 4k), suggesting that RPL7L does not get incorporated into ribosomes in these cells. Taken together, these results characterize the direct interactomes of 18S, 5.8S and 28S rRNAs in living cells, highlighting the robustness of TREX for comprehensive profiling of distinct rRNA–protein interactions.

### TREX reveals the interactors of 5′-ETS, ITS1, ITS2 and 3′-ETS

Having revealed the direct binders of 18S, 5.8S and 28S rRNA segments, we next interrogated the 5′-ETS (~3.6 kb), ITS1 (~1.1 kb), ITS2 (~1.2 kb) and 3′-ETS (361 b) spacer regions (Fig. 5a). As with other regions of 45S rRNA, we validated the efficiency of each region-specific depletion through RT–qPCR (Extended Data Fig. 5a–d). Subsequently, we conducted quantitative MS analysis of the TREX samples, originating from

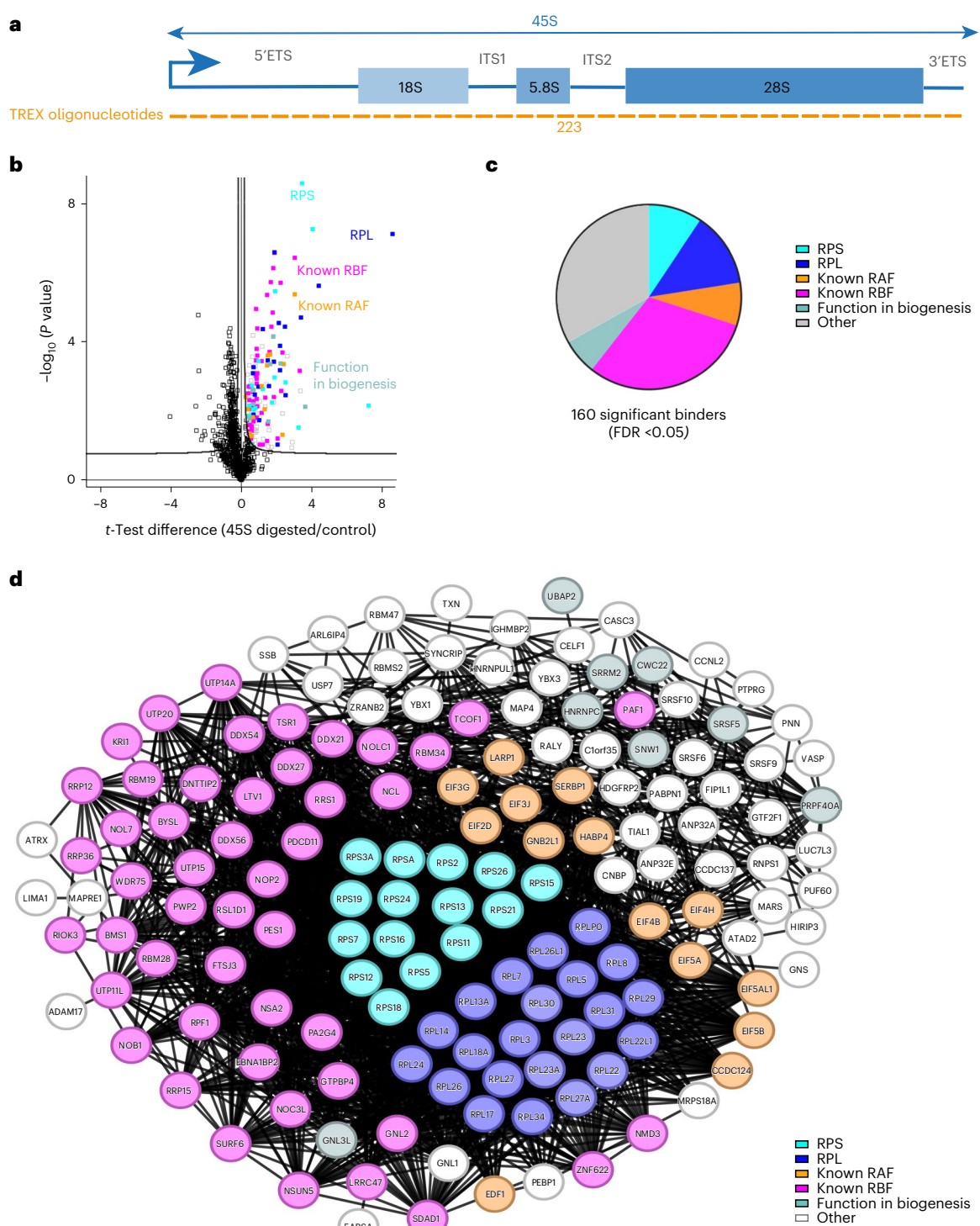

**Fig. 3 | TREX captures the compendium of proteins that bind to the 45S rRNA. a**, Schematic representation of 45S rRNA primary sequence and its various segments. The tiling antisense DNA oligonucleotides used for depletion in TREX (223 in total) are depicted on the graph. **b**, Volcano plot of the two-sided two-sample *t*-test comparison of 45S rRNA digested versus undigested TREX samples (*n* = 4 biological replicates), showing the enrichment of RPS (cyan) and RPL (blue) proteins, known RBFs (purple)[38] and known RAFs (orange), as well as previously unknown interactors suspected of involvement in human ribosome biogenesis based on the impact of their depletion (teal)[39–41]. Curved lines mark the significance boundary (FDR = 0.05, S0 = 0.1). **c**, Pie diagram of the composition of proteins identified in the 45S rRNA TREX. Proteins are color-coded as described in **b**. **d**, Interaction network analysis of 45S-bound proteins identified by TREX, using the STRING physical interactions database[42]. Proteins are color-coded as described in **b**.

ten million HCT116 cells per replicate experiment. TREX analysis of the 5′-ETS revealed a predominant interaction with known RBFs involved in early pre-rRNA processing[38] (Fig. 5b,c and Supplementary Dataset 7). A number of other key interactors of the 5′-ETS, such as NCL[46] and DNTTIP2 (ref. 47), were also identified (Fig. 5b and Supplementary Dataset 7). As expected, ribosomal proteins were not prominently represented among the interactors of 5′-ETS. Only RPS9, RPS25 and RPL23A were detected as significant 5′-ETS binding partners (Fig. 5b and

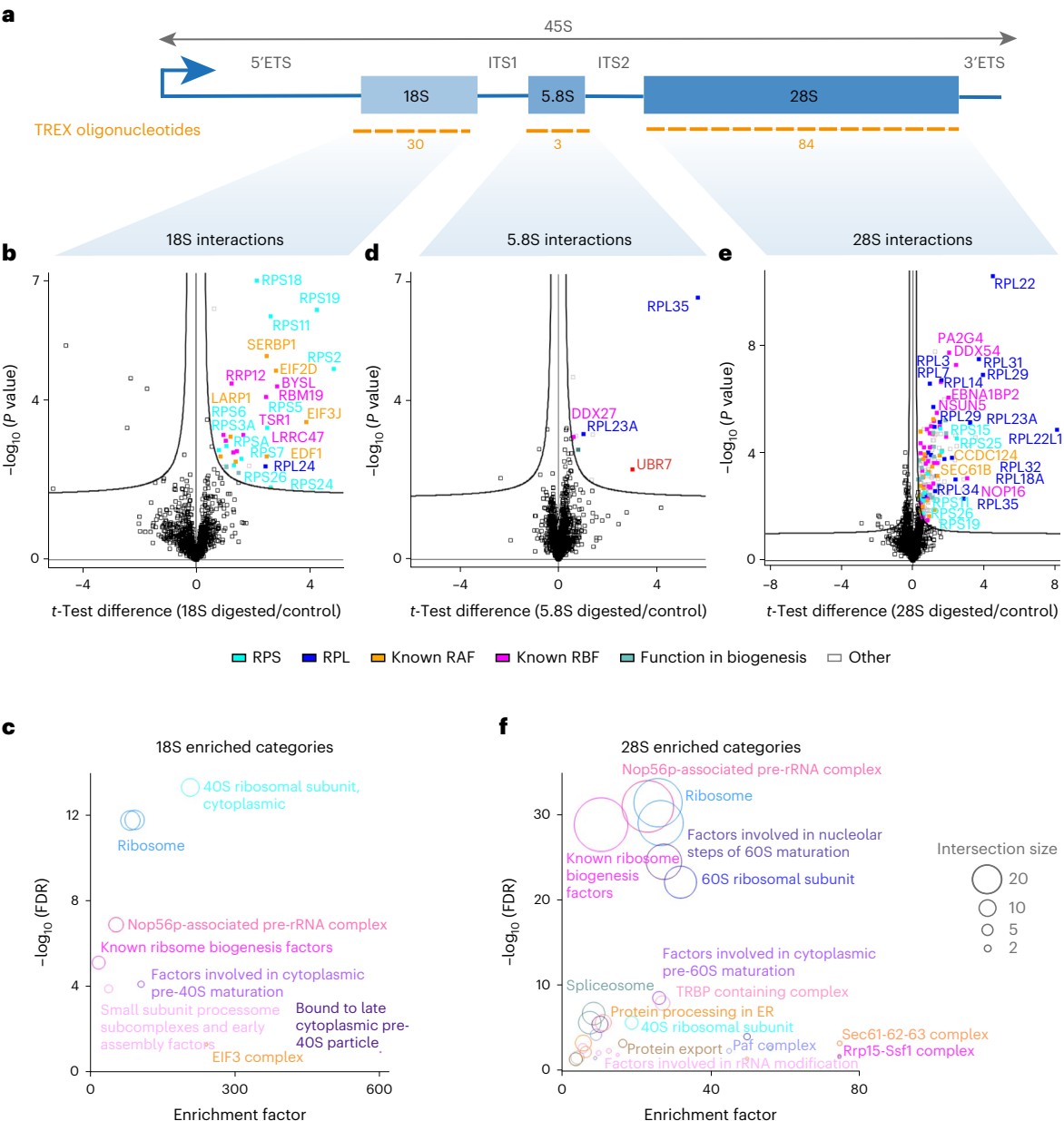

**Fig. 4 | TREX defines the interactomes of 18S, 5.8S and 28S rRNA. a**, Schematic representation of the 18S, 5.8S and 28S segments within the 45S rRNA. The tiling antisense DNA oligonucleotides used for depletion of each segment in each set of TREX experiments are depicted on the graph. **b**, Volcano plot of the two-sided two-sample *t*-test comparison of 18S rRNA digested versus undigested TREX samples (*n* = 4 biological replicates), showing the enrichment of RPS proteins, known small subunit RAFs and known small subunit processing RBFs, as well as RPL24. **c**, Fisher's exact test analysis of known protein categories that are over-represented among the 18S interactors (Benjamini–Hochberg FDR < 0.05). Each circle represents an enriched category extracted from Dorner et al.[38] or the Kyoto Encyclopedia of Genes and Genomes (KEGG) database[68], with circle size representing the number of shared proteins. **d**, Volcano plot of the two-sided

two-sample *t*-test comparison of 5.8S rRNA digested versus undigested TREX samples (*n* = 5 biological replicates), showing the enrichment of DDX27, RPL23A, RPL35 and UBR7. **e**, Volcano plot of the two-sided two-sample *t*-test comparison of 28S rRNA digested versus undigested TREX samples (*n* = 5 biological replicates), showing the enrichment of RPL proteins, known large subunit RAFs and known large subunit processing RBFs, as well as RPS11, RPS15, RPS19, RPS25 and RPS26. Curved lines in all volcano plots mark the significance boundary (FDR = 0.05, S0 = 0.1). **f**, Fisher's exact test analysis of known protein categories that are over-represented among the 28S interactors (Benjamini–Hochberg FDR < 0.05). Each circle represents an enriched category extracted from Dorner et al.[38] or the KEGG database, with circle size representing the number of shared proteins. ER, endoplasmic reticulum.

Supplementary Dataset 7). RPS25 and RPL23A, along with RPS17, were also found to interact with several other spacer regions (see below), indicative of their potential roles in ribosome biogenesis beyond ribosome constituents. In contrast, RPS9 specifically emerged as a 5′-ETS interactor. Inspection of the human 80S ribosome structure[43] revealed a direct association of RPS9 with the 5′ end of 18S rRNA, where the junction between the 5′-ETS and 18S (cleavage site 1) once stood (Extended Data Fig. 5e). Consequently, the release of RPS9 in 5′-ETS TREX may be

attributed to its binding across this junction site, which is covered by our 5′-ETS tiling oligonucleotides.

TREX analysis of ITS1 identified distinct RBFs primarily involved in ITS1 processing[38] (Fig. 5d,e and Supplementary Dataset 8). The most prominently enriched protein was RPP25L, a paralogue of RPP25 that constitutes a subunit of the RMRP endonuclease complex[48] (Fig. 5d and Supplementary Dataset 8). This complex facilitates the critical site 2 cleavage within ITS1, which separates the pre-40S and pre-60S

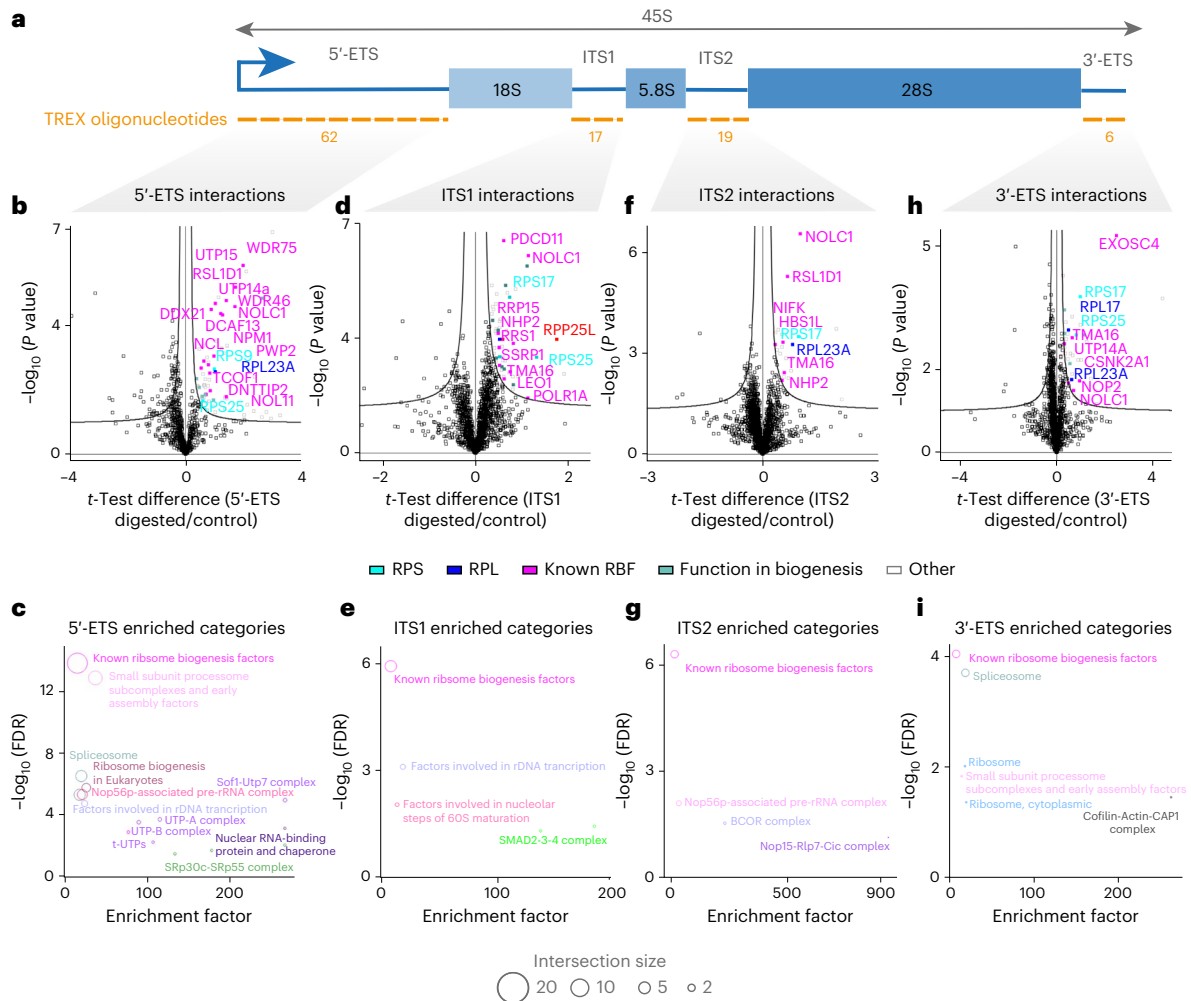

**Fig. 5 | TREX defines the interactomes of 5'-ETS, ITS1, ITS2 and 3'-ETS spacer regions. a**, Schematic representation of the 5'-ETS, ITS1, ITS2 and 3'-ETS segments within the 45S rRNA. The tiling antisense DNA oligonucleotides used for depletion of each segment are depicted on the graph. **b**, Volcano plot of the two-sided two-sample *t*-test comparison of 5'-ETS digested versus undigested TREX samples (*n* = 5 biological replicates) showing the prominent enrichment of several known early processing RBFs, along with RPS9, RPS25 and RPL23A. **c**, Fisher's exact test analysis of known protein categories that are over-represented among the 5'-ETS-interacting proteins (Benjamini–Hochberg FDR < 0.05). Each circle represents an enriched category extracted from Dorner et al.[38] or the KEGG database, with circle size representing the number of shared proteins. **d**, Volcano plot of the two-sided two-sample *t*-test comparison of ITS1 digested versus undigested TREX samples (*n* = 5 biological replicates), showing the enrichment of various RBFs, RPS17, RPS25 and RPP25L. **e**, Fisher's exact test analysis of known protein categories that are over-represented among the ITS1 interacting proteins (Benjamini–Hochberg FDR < 0.05). Each circle represents an enriched category extracted from Dorner et al.[38] or the KEGG database, with circle size representing the number of shared proteins. **f**, Volcano plot of the two-sided two-sample *t*-test comparison of ITS2 digested versus undigested TREX samples (*n* = four biological replicates), showing the enrichment of several known RBFs, RPS17 and RPL23A. **g**, Fisher's exact test analysis of known protein categories that are over-represented among the ITS2 interacting proteins (Benjamini–Hochberg FDR < 0.05). Each circle represents an enriched category extracted from Dorner et al.[38] or the KEGG database, with circle size representing the number of shared proteins. **h**, Volcano plot of the two-sided two-sample *t*-test comparison of 3'-ETS digested versus undigested TREX samples (*n* = 4 biological replicates), showing the enrichment of EXOSC4 and several known RBFs, along with RPS17, RPS25, RPL23A and RPL17. Curved lines in all volcano plots mark the significance boundary (FDR = 0.05, S0 = 0.1). **i**, Fisher's exact test analysis of known protein categories that are over-represented among the 3'-ETS-interacting proteins (Benjamini–Hochberg FDR < 0.05). Each circle represents an enriched category extracted from Dorner et al.[38] or the KEGG database, with circle size reflecting the number of shared proteins.

processing particles[49]. Interestingly, the protein expression level of RPP25L in HCT116 cells is over 20-fold higher than that of RPP25 (Extended Data Fig. 5f), suggesting a likely expression-based switch between the two paralogs in these cells. To assess a role for RPP25L in ribosome biogenesis of HCT116 cells, we depleted it using RNAi, and examined the steady-state levels of ITS1, 5'-ETS and ITS2 by RT–qPCR. RPP25L depletion led to a significant accumulation of ITS1, while 5'-ETS and ITS2 levels remained unchanged (Extended Data Fig. 5g,h), indicating a specific disruption in ITS1 processing. These findings establish RPP25L as a previously unknown RBF that is involved in ITS1 processing.

Additionally, TREX revealed an interaction between ITS1 and SMAD proteins (SMAD2, 3 and 9), which are traditionally associated with TGFB signaling (Fig. 5e and Supplementary Dataset 8). This discovery aligns well with a recent report that unveiled an unanticipated role for *Drosophila* smad2 in regulating ITS1 cleavage[50], thereby supporting a direct association between SMADs and ITS1 that is conserved from *Drosophila* to humans. Among the direct interactors of ITS2, we found NIFK, the mammalian homolog of Nop15, which plays a key role in ITS2 folding (Fig. 5f,g and Supplementary Dataset 9). Additionally, TMA16 and NHP2, two RBFs involved in pre-60S processing and

modification[51,52], were found as direct binders of both ITS regions (Fig. 5d,f and Supplementary Datasets 8 and 9).

Finally, we examined the interactome of 3′-ETS—a region that has remained poorly characterized compared with other spacer segments. EXOSC4, a core subunit of the exosome complex, was identified as one of the top interactors (Fig. 5h and Supplementary Dataset 10). This aligns with recent evidence suggesting that 3′-ETS-containing pre-rRNAs can recruit the exosome complex for pre-rRNA surveillance[53]. Several RBFs were also identified as direct interactors of 3′-ETS. These included certain early processing factors (Fig. 5h,i and Supplementary Dataset 10), supporting the hypothesis that some 5′-ETS and 3′-ETS processing events may be coupled in human cells[54]. Similar to other spacer regions, ribosomal proteins were not prominently identified, with the exception of a select number that were still present. Of these, RPS17, RPS25 and RPL23A were also identified as interactors of several other spacer regions, whereas RPL17 exhibited specificity for the 3′-ETS (Fig. 5h and Supplementary Dataset 10). Similar to the case of RPS9 and the 5′-ETS, we could demonstrate that the direct contact site for RPL17 in the human 80S ribosome structure is formed at the 3′ end of 28S rRNA (cleavage site 02), where the junction with the 3′-ETS was once located (Extended Data Fig. 5i). Consequently, the release of RPL17 in our TREX analysis of 3′-ETS may be attributed to its binding across this junction site before its cleavage. Collectively, these findings provide a comprehensive catalog of the direct interactions of each transcribed spacer region in human 45S rRNA.

### TREX reveals region-specific and multiregional rRNA binders

Next, we set out to integrate the TREX results from different 45S rRNA segments, to map the positional landscape of RNA–protein interactions across the full length of 45S pre-rRNA. For this purpose, we selected proteins exhibiting significant interactions with at least one segment of 45S rRNA, and conducted unsupervised hierarchical clustering of the *t*-test scores across the different TREX experiments. Remarkably, more than 95% of the interacting proteins (378 out of 396) were found to be bound primarily to a single region of 45S rRNA, forming distinct region-specific clusters (Fig. 6a and Supplementary Dataset 11). Clusters 1 and 2 were comprised of the 18S interactors, encompassing RPSs and small subunit RBFs. Cluster 3 consisted of interactors specific to the 5′-ETS, prominently featuring members of different UTP complexes. Interactors specific to ITS1 were found in cluster 4, featuring known participants such as PDCD11 and RPP25L. Cluster 5 contained interactors of both the 5.8S and ITS2 regions, featuring established partners such as NIFK. Clusters 6 and 7 constituted primarily interactors specific to the 3′-ETS, while clusters 8 and 10, comprising the largest group of region-specific interactors, were associated predominantly with the 28S segment, containing many large subunit RBFs and RPLs (Fig. 6a and Supplementary Dataset 11). RPL35 was the only protein that did not cluster with any other interactor, as it exhibited a unique profile of primary association with 5.8S with some binding also to 28S (Supplementary Dataset 11)—a pattern consistent with its position in the human 80S ribosome structure (Extended Data Fig. 4e).

In contrast, cluster 9 exhibited enrichment across several regions (Fig. 6a and Supplementary Dataset 11), suggesting a distinct mode of interaction. This cluster contained 18 proteins, including NOLC1 and TCOF1, which form a complex that couples rDNA transcription with the rRNA modification machinery[55], as well as several splicing factors (Fig. 6b,c and Supplementary Dataset 11). Previous studies have implicated certain serine/arginine (SR)-rich splicing factors such as SRSF1 and PRP43 in regulation of pre-rRNA-related processes[56–58]. However, our findings suggest a broader involvement of splicing factors in ribosome biogenesis than previously known. In agreement with this notion, five members of this cluster have been identified in large-scale RNAi screens for human ribosome biogenesis regulators[39–41] (Fig. 6d). Moreover, eight members can localize to the nucleolus, according to the Human Protein Atlas[59], further supporting an involvement in ribosome

biogenesis (Fig. 6d). We also found existing eCLIP data for three members of this cluster (SRSF9, GTF2F1 and NOLC1)[5,6,60]. Re-analysis of this data by mapping the reads to the *45S* rDNA locus, which is often excluded in standard CLIP analyses, revealed numerous binding sites across the full length of 45S (Extended Data Fig. 6a)—a binding pattern consistent with our TREX analysis. Based on this pattern of interaction, we hypothesized that, rather than being involved in a specific step of rRNA processing, members of cluster 9 may have a role in regulating rRNA synthesis. To assess this hypothesis, we used RNAi to deplete ten proteins in the cluster 9 that were not previously implicated in ribosome biogenesis, and assessed the impact on rRNA synthesis by nascent RNA imaging[61]. A significant reduction in nascent rRNA levels was detected upon depletion of 9 out of 10 proteins (Extended Data Fig. 6b,c). Collectively, our cross-comparison depicts a specific positional binding pattern for most 45S-associated proteins, but we also identify a distinct cluster of proteins encompassing several splicing factors that interact with several pre-rRNA regions, and probably function in promoting rRNA synthesis.

## Discussion

In this study, we present TREX, a highly efficient and specific RNA-centric method that enables unbiased region-specific exploration of direct RNA–protein interactions in their endogenous context. The power of TREX lies in its combination of phase extraction and RNase H-mediated RNA degradation (Fig. 1a). Through phase extraction, TREX effectively isolates RNA-bound from nonbound proteins. This, combined with RNase H-mediated degradation, enables the precise and stoichiometric release and recovery of proteins that are bound to a given RNA sequence. Currently, the only other available RNA-centric approach for assessing region-specific RNA–RBP interactions in cells is incPRINT, which uses a reporter-based system where tagged RBPs and RNA sequences of interest are ectopically coexpressed in a cell line, followed by cell lysis and assessment of RNA–RBP copurification via a luminescence based reporter assay[62]. In comparison, TREX is unbiased rather than candidate based, does not require any genetic manipulations and enables the endogenous interactome profiling of target sequence with minimal preparation. An additional advantage of TREX lies in its cost-effectiveness, as it requires only unlabeled tiling antisense DNA oligonucleotides, along with generic reagents commonly used in most RNA biology laboratories. This affordability substantially enhances the accessibility of TREX to researchers worldwide.

RNA affinity capture is the approach used most commonly for probing direct interactors of a given RNA target under endogenous settings[1]. However, this approach is unsuited to region-specific interaction analysis, as target RNA molecules can only be purified in their entirety through affinity capture. TREX addresses this shortcoming by using the robust enzymatic reaction of RNase H, which enables near complete region-specific target degradation, with phase extraction providing an immaculate system to separate the RNA-bound from the released RBPs. To assess the performance of TREX in comparison with RNA affinity capture-based methods, we conducted benchmarking experiments using the well-characterized U1 snRNA as the target. Our results demonstrated the superior performance of TREX compared with three previously published affinity capture-based studies[9,10,12]. We also demonstrated the ability of TREX to investigate region-specific interactions by applying it to reveal the direct binding partners of the most conserved segment of NORAD lncRNA[27,28]. Our analysis showed a substantial overlap between TREX and previous NORAD interactome capture studies, revealing both known and previously unknown direct interactions. We also estimated the relative affinities of the identified binding partners by comparing their absolute protein abundance in TREX samples relative to that of the total cell extracts, a feat only possible in TREX thanks to the stoichiometric recovery of RNA-bound proteins. Such affinity estimations can be extremely useful for revealing

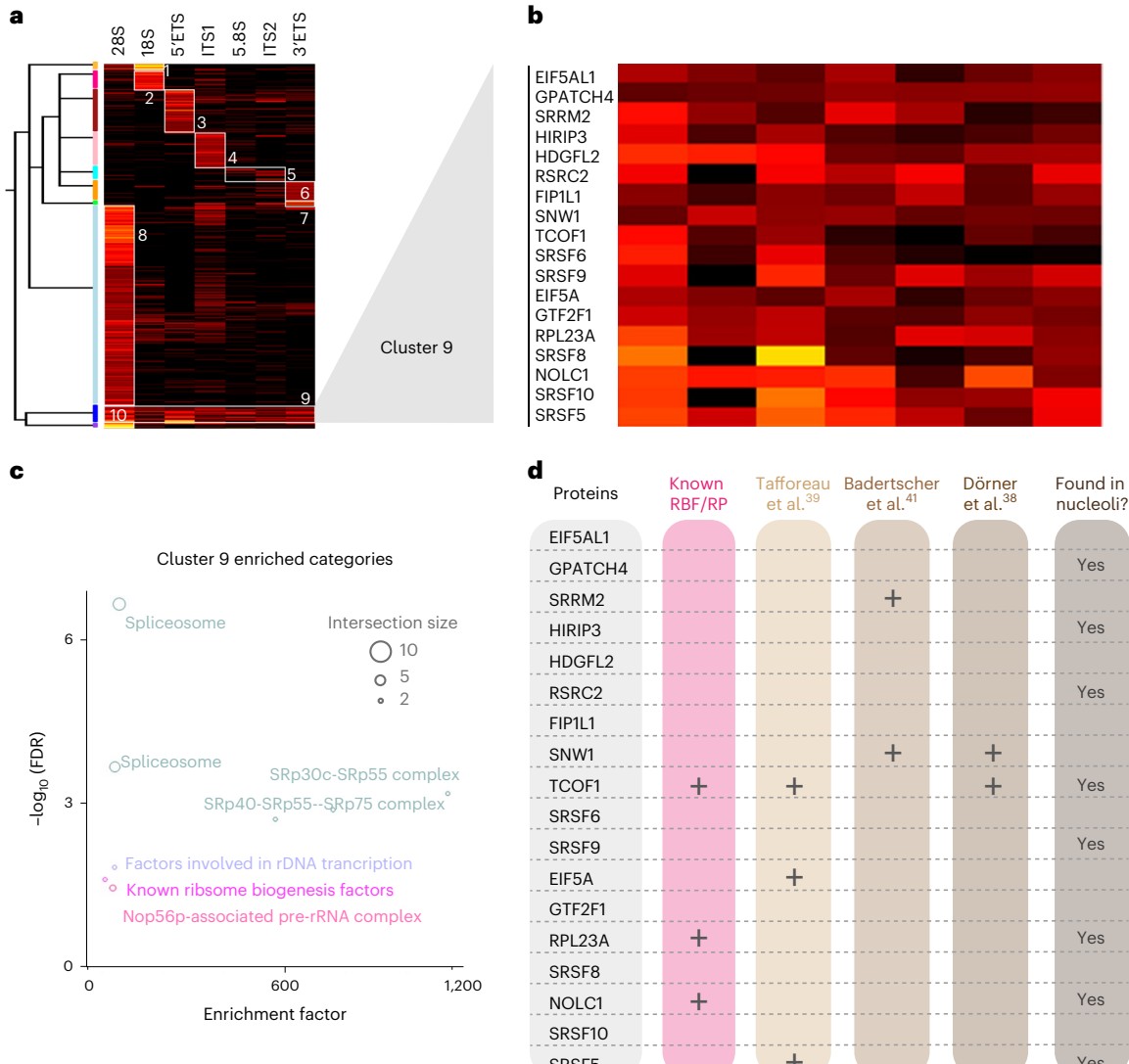

**Fig. 6 | TREX reveals region-specific and multiregional interactors of 45S rRNA. a**, Unsupervised hierarchical clustering analysis of *t*-scores for significant hits from TREX analyses of different segments of 45S rRNA, with complete Euclidean distance calculation and K-means preprocessing (*t*-score color scale, black, 0 or less; red, 2–6; orange, 6–10; yellow, 10 or more). **b**, Zoomed-in view of cluster 9 and the list of its constituent proteins. **c**, Fisher's exact test analysis of known protein categories that are over-represented among the cluster 9 proteins (Benjamini–Hochberg FDR < 0.05). Each circle represents an enriched category extracted from Dorner et al.[38] or the KEGG database, with circle size representing the number of shared proteins. **d**, Overlap analysis of cluster 9 proteins with lists of known RBFs from ref. 38 and hits from three large-scale RNAi screens of human ribosome biogenesis regulators[39–41]. Proteins with reported localization to the nucleolus, either as primary or as an additional site of localization, according to the Human Protein Atlas, are also marked on the graph.

protein candidates of 'ribo-regulation'[63], and will assist researchers with prioritization of hits for downstream analyses.

Leveraging the power of TREX for positional mapping of RNA–protein interactions, we generated a comprehensive region-by-region interactome map of human 45S rRNA. This revealed a highly specific interactome landscape for each segment of 45S rRNA, consisting of ribosomal proteins, RBFs, RAFs and previously unknown rRNA-binding proteins, some of which were subsequently validated as ribosome biogenesis regulators. Remarkably, while most of the identified proteins exhibited highly specific regional binding patterns, a cohort of 18 proteins was found to interact with several 45S rRNA regions. Several splicing factors were present among this group, some of which were previously shown to impact ribosome biogenesis[39–41]. Systematic RNAi depletion revealed a role for many members of this cohort in promoting rRNA synthesis, but their exact mechanisms of action remain to be determined.

As nascent 45S rRNA emerges from RNAPI, it becomes decorated by several ribosomal proteins and RBFs, giving rise to a distinct nucleolar particle known as the 90S pre-ribosome. Through a series of endonucleolytic cleavages and exonucleolytic processing events, along with rRNA modifications, folding and assembly of additional ribosomal proteins, RBFs and 5S rRNA, the 90S particle splits into pre-40S and pre-60S particles and gradually matures into translationally competent small and large ribosomal subunits[38]. Recent advances in determining the composition and structures of ribosome intermediates have relied primarily on affinity purification of selected RBFs that associate with specific stages of ribosome biogenesis[64,65]. Consequently, the emerging understanding of ribosome biogenesis from such studies is primarily protein-centric, with transient interactions that may not be particularly amenable to survive protein affinity purifications likely to be under-represented or lost. Our TREX analysis provides an alternative RNA-centric view of the proteins that associate with each segment of

45S rRNA, revealing potentially new ribosome biogenesis components that could have been missed. Another important limitation in our current understanding comes from ignoring the considerable allelic variation that is present within the rRNA sequences of higher eukaryotes, which are known to have notable biological ramifications[66,67]. TREX, due to its sequence-specific nature, can be adapted to study these variations, thus shedding light on their impact on ribosome biogenesis and function.

## Online content

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

## Methods

Full details of the TREX tiling antisense DNA oligonucleotides (Supplementary Table 1), the RT–qPCR probes (Supplementary Table 2) and all other reagents and materials used in this study (Supplementary Table 3) are available as additional supplementary information. For a detailed step-by-step protocol of TREX, please visit https://www.mardakhehlab.info/resources/trex.

### Cell culture

HCT116 cells were cultured in McCoy's 5a medium supplemented with 10% fetal bovine serum and 100 U of penicillin/0.1 mg ml$^{-1}$ streptomycin. The cells were maintained at 37 °C in a humidified incubator with 5% $CO_2$. HCT116 were examined routinely for mycoplasma contamination and their identity was confirmed by short tandem repeat profiling (date of the last profiling: July 2023).

### RNAi depletion

A total of 50,000 HCT116 cells were seeded per well of a 12-well plate and grown overnight. The following day, the cells were transfected with 10 nM of non-targeting control or gene-specific siRNA smartpools, using Lipofectamine RNAi MAX, according to the manufacturer's instructions. The cells were maintained for a further 72 h to achieve sufficient depletion, before collection by trypsinization for downstream analyses.

### Targeted RNase H-mediated extraction of crosslinked RBPs

HCT116 cells were seeded on 150 mm tissue culture-treated culture dishes and allowed to attach for a minimum of 16 h, before being washed with ice-cold PBS and irradiated on ice with 200 mJ cm$^{-2}$ of UV-C (254 nm), using a Hoefer Scientific UV Crosslinker. Cells were then lysed by direct addition of TRIZOL to each dish (1 ml per ~20 million cells). We used 100 million cells per biological replicate experiment for NORAD ND4, 30 million cells per replicate for full-length 45S rRNA, and 10 million cells per replicate for all other TREX experiments. After scraping the cells in TRIZOL, homogenized lysates were incubated for 5 min at room temperature to dissociate noncrosslinked RNA–protein interactions, before the addition of chloroform (200 μl per 1 ml of lysate) and centrifugation for 15 min at 12,000$g$ at 4 °C to induce phase separation. The aqueous and organic phases were subsequently discarded, and the interface was resolubilized in TRIZOL. Phase separation and resolubilization was repeated two more times to remove any noncrosslinked RNA or proteins. The isolated interface was then washed gently with TE buffer to remove any traces of TRIZOL, before being solubilized by resuspension in TE buffer supplemented with increasing concentrations of SDS, as described previously[20]. The RNA–protein crosslinks were then precipitated by adding NaCl (to 300 mM final concentration) followed by isopropanol (to 50% final concentration), and the pellet was washed with 70% ethanol and resuspended in nuclease-free water. Any contaminating genomic DNA was removed by treating the samples with TURBO DNase (2 U per million cells) for 50 min at 37 °C, followed by a further round of isopropanol precipitation. The pellets were washed again with 70% ethanol and resuspended in probe hybridization buffer (50 mM NaCl, 1 mM EDTA, 100 mM Tris-HCl, pH 7.0), before a pool of tiling antisense DNA oligonucleotides complementary to the target RNA region of interest was added to the mixture. The individual oligonucleotides used were non-overlapping, on average 60 nt long (and no shorter than 30 or longer than 90 nt) and unmodified. We ensured that no oligonucleotides strongly matched to an off-target RNA by blasting their sequences against the RefSeq database. The sequences of all TREX antisense tiling probes used in this study can be found in Supplementary Table 1. The amount of oligonucleotide pool used per reaction was determined before the TREX experiment by performing oligonucleotide dose titration for each target, followed by RT–qPCR analysis (Extended Data Fig. 1). Annealing was performed in a Thermomixer by first heating the samples to 95 °C for 2 min to denature

all RNAs, followed by a gradient temperature drop of 2 °C min$^{-1}$, until the samples reached 50 °C. DNA-hybridized RNA regions were subsequently digested following the addition of MgCl$_2$ to neutralize the EDTA, followed by Thermostable RNase H (1 U per million cells) to each reaction and further incubation at 50 °C for 60 min. As controls, samples were treated identically but no RNase H enzyme was added. Samples were then centrifuged at 12,000$g$ for 3 min and the supernatant was transferred to a new tube. Then, 10% of each supernatant was aliquoted and taken for RNA extraction to analyze target depletion efficiency and specificity (see below), while the remaining 90% was subjected to a second round of organic phase separation by adding 900 μl of TRIZOL LS to each sample, followed by 200 μl of chloroform, mixing and centrifugation for 15 min at 12,000$g$ at 4 °C. The released RBPs were recovered by taking the organic phase into a new tube. Ice-cold acetone was added to each sample (final concentration of 80%), followed by mixing and overnight incubation at −20 °C to precipitate the released RBPs. The pellets were subsequently recovered and washed with 1 ml of 80% acetone, followed by centrifugation for 15 min at 16,000$g$ at 4 °C. This step was repeated once, and the final pellet was air-dried for 5 min at room temperature.

### Total lysate preparation

For label-free absolute protein quantification analysis of the HCT116 proteome, 500 μl aliquots of TRIZOL-lysed UV-C crosslinked HCT116 cells (corresponding to ten million cells) before TREX analysis were taken and subjected to protein precipitation by adding ice-cold acetone to the final concentration of 80%, followed by mixing and overnight incubation at −20 °C. The precipitated protein pellets were subsequently recovered and washed with 1 ml of 80% acetone, followed by centrifugation for 15 min at 16,000$g$ at 4 °C. This was repeated once, and the final pellet was air-dried for 5 min at room temperature.

### MS sample preparation and data acquisition

Acetone-precipitated proteins from TREX were subjected to in-solution digestion by trypsin as described previously[61]. Briefly, proteins were recovered in 200 μl of 2 M urea, 50 mM ammonium bicarbonate and reduced by adding dithiothreitol to a final concentration of 10 mM. After 30 min of incubation at room temperature, samples were alkylated by adding 55 mM iodoacetamide and incubated for another 30 min at room temperature in the dark. Trypsin digestion was then performed using 2 μg of trypsin per sample. The next day, samples were desalted using the Stage Tip procedure[69], and recovered in 0.1% trifluoroacetate, 0.5% acetic acid, 2% acetonitrile (A* buffer) for liquid chromatography with tandem MS (LC–MS/MS) analysis. For total proteomics analysis, samples were similarly dissolved in 200 μl of 2 M urea, 50 mM ammonium bicarbonate (ABC), reduced and alkylated with dithiothreitol and iodoacetamide, followed by trypsin digestion of an equivalent of ~100 μg of protein, using the FASP protocol[70]. The digested peptides were subjected to peptide fractionation using a Pierce high pH reverse-phase fractionation kit, as described previously[71]. Fractions were dried with vacuum centrifugation before recovery in A* buffer for LC–MS/MS analysis. LC–MS/MS analysis was performed as described before[61], using a Q Exactive-plus Orbitrap mass spectrometer coupled with a nanoflow ultimate 3000 RSL nano HPLC platform (ThermoFisher). The instrument was operated using the Thermo Xcalibur v.4.5 SP1 software. For total proteomics analysis, the equivalent of ~1 g of total protein per fraction was injected into the instrument, while ~90% of the total peptide mixture per each TREX experiment was injected. Samples were resolved at flow rate of 250 nl min$^{-1}$ on an Easy-Spray 50 cm × 75 m RSLC C18 column (ThermoFisher), using a 123 min gradient of 3% to 35 % of Buffer B (0.1% formic acid in acetonitrile) against Buffer A (0.1% formic acid in LC–MS gradient water). LC-separated samples were infused into the MS by electrospray ionization. Spray voltage was set at 1.95 kV, and capillary temperature was set to 255 °C. MS was operated in data dependent

positive mode, with 1 MS scan followed by 15 MS2 scans (top 15 method). Full scan survey spectra ($m/z$ 375–1,500) were acquired with a 70,000 resolution for MS scans and 17,500 for the MS2 scans. A 30 s dynamic exclusion was applied to all runs.

## MS data analysis

Maxquant (v.1.6.3.3) was used for all MS searches and quantifications[72]. Raw data files were searched against a FASTA file of the human proteome, excluding variants and isoforms, extracted from UNIPROT. Enzyme specificity was set to 'Trypsin,' allowing up to two missed cleavages. Label-free quantification was enabled using label-free quantification intensity calculation with a minimum ratio count of two. Variable modifications were set at oxidation (M), acetylation (N-term) and phosphorylation (ST). False discovery rates (FDR) were calculated using a reverse database search approach, and were set at 1% for identification of peptides, modifications and proteins. 'Match between runs,' 'Re-quantify' and 'iBAQ' options were enabled. Default Maxquant parameters were used for all other settings. All downstream data analyses such as data filtering, log transformation, imputation of missing values (downshift, 1.8 s.d.; variation, 0.3 s.d.), two-sample $t$-test analysis, category enrichment analysis and hierarchical clustering, were performed by Perseus (v.1.6.2.3)[73]. PCA was performed on transformed and imputed data. Where one TREX sample behaved overtly differently to the other three or four biological replicates based on the PCA scattering, the outlier was subsequently removed from downstream analysis. For two-way two-sample $t$-test analyses, permutation-based $P$ value correction with an FDR cut-off of <0.05 and an artificial within-groups standard deviation (S0) of 0.1 was used. Category enrichment analysis was performed using Fisher's exact test, with a Benjamini–Hochberg FDR cut-off of <0.02. Clustering of $t$-scores, combined from different TREX $t$-test analyses, was performed using Euclidean complete distance calculation, with K-means preprocessing. Only proteins that were identified as significant binders in at least one TREX experiment were included for clustering analysis.

## RNA affinity estimations

The affinity of the TREX-identified proteins to their target RNA was estimated by calculating the relative ratio values of RNA-bound to total iBAQ protein abundance levels as follows:

$$\text{Estimated affinity (Ka)} = \frac{[\text{RNA bound}]}{[\text{Total}]} = \frac{(\text{iBAQ TREX})}{(\text{iBAQ total})}.$$

The iBAQ TREX value for each hit was calculated by subtracting the averaged iBAQ values form the +RNase H TREX Maxquant runs by the averaged iBAQ values from the −RNase H TREX runs. The iBAQ total value for each hit was simply extracted from the total cell lysate Maxquant analysis. The relative affinity was then calculated for all the identified hits and displayed as a ranked plot.

## RNA isolation, cDNA synthesis and qPCR

Target depletion efficiency upon RNase H treatment was determined as follows: an aliquot of solubilized and annealed interface, with or without RNase H treatment, corresponding to around two million cells, was digested with 200 µg of proteinase K in 200 µl of 10 mM Tris pH 8.0, 1 mM EDTA, 0.5% SDS for 1 h at 56 °C. RNA was then extracted with TRIZOL LS, following the manufacturers' instructions, before cDNA synthesis and quantitative real-time PCR analysis in a single reaction, using the Power SYBR Green RNA-to-CT 1-Step kit. Assessment of RNA levels from siRNA-treated cells was done as follows: HCT116 cells were trypsinized and collected by centrifugation for 5 min at 300$g$. Cell pellets were then lysed in TRIZOL and RNA extracted according to manufacturer's instructions. The purified RNA was then subjected to cDNA synthesis and quantitative real-time PCR analysis in a single reaction, using the Power SYBR Green RNA-to-CT 1-Step kit. The Quant-Studio 7 Flex real-time PCR system (Applied Biosystems) with the following program was employed for qPCR analysis: 48 °C for 30 min,

95 °C for 10 min followed by 40 cycles of 95 °C for 15 s and 60 °C for 60 s. For input normalization, GAPDH and RPS18 were used as housekeeping transcripts, except for the 45S region-specific experiments, where 18S was used instead of RPS18, to provide a further control for the region-specific depletions (with the exception of 18S TREX itself). RNA expression levels were estimated using $2^{-\Delta CT}$. For ITS2, the qPCR probes used led to amplification of an additional nonspecific peak with a lower melting temperature, so to delineate the specific peak from the nonspecific one, relative area under the curve for the higher melting peak was measured specifically using the DescTools package in R (v.0.99.52). The sequences of all qPCR primers used in this study are provided in Supplementary Table 2.

## Crosslinking and immunoprecipitation

CLIP was performed as described previously[61], with some modifications. Briefly, HCT116 cells were irradiated on ice with 150 mJ cm² UV light (254 nm) in PBS, using a Hoefer Scientific UV crosslinker. Cells were subsequently collected by scraping and centrifugation for 5 min at 300$g$. Pellets were lysed in RNase-free CLIP lysis buffer (50 mM Tris-HCl pH 7.5, 100 mM NaCl, 1% Igepal CA-630, 0.1% SDS, 0.5% sodium deoxycholate, supplemented with Roche mini protease inhibitors tablets), sonicated in a sonicator bath (Bioruptor Pico) for ten cycles (30 s on, 30 s off), and cleared by centrifugation (12,000$g$, 10 min). The protein concentrations of the cleared lysates were measured and balanced to 1 mg ml⁻¹. RNA was subsequently subjected to partial digestion with 0.2 U ml⁻¹ of RNase I, along with removal of contaminating DNA by co-addition of 4.0 U ml⁻¹ of TURBO Dnase and incubation at 37 °C for 3 min. The lysates were supplemented with SUPERase-In RNase inhibitor, and 10% aliquots of each sample were collected for input RNA and protein (western blotting) analysis. The remaining lysates were subjected to immunoprecipitation by incubation of ~1 mg of each sample with ~5 µg of antibody, preconjugated to 50 µl of beads (Dynabeads Protein G). The mixture was incubated for 1 h (or overnight depending on the antibody used) at 4 °C on a rotating wheel. The beads were washed three times with 0.9 ml of CLIP lysis buffer. On the last wash, a 10% aliquot was taken from each immunoprecipitation for protein analysis. Beads and half of the input material taken for RNA analysis were subsequently supplemented with 0.2 ml of CLIP elution buffer (50 mM Tris-HCl, pH 7.5, 100 mM NaCl, 1% Igepal CA-630, 1% SDS, 0.5% sodium deoxycholate) and digested by addition of 7 µl of RNA Grade Proteinase K (20 mg ml⁻¹) and incubating at 65 °C for 1 h with mixing. The RNA was extracted by adding 0.75 ml TRIZOL LS to each sample, followed by 200 µl of chloroform. Samples were centrifuged at 12,000$g$ for 15 min at 4 °C and the aqueous phase was isolated and precipitated with one volume of isopropanol and 1/10 volume of sodium acetate overnight at −80 °C. The RNA pellets were washed twice with 70% ethanol, then dried and resuspended in 15 µl of RNAse-free water. All samples were diluted 1:1 with RNAse-free water and 2 µl were used as input for quantitative real-time RT–qPCR analysis. In parallel, the aliquots of input and immunoprecipitate that were taken for protein analyses were resuspended in lithium dodecyl sulfate sample buffer and subjected to western blotting with the indicated antibodies, as described previously[61].

## Nascent RNA imaging

Nascent RNA imaging, to quantify newly synthesized rRNA levels, was done as described before[61]. Briefly, cells were seeded on 18-well Flat µ-Slides from iBidi and grown for 24 h, before being pulsed for 30 min with 2 mM 5-fluorouridine (FUrd) to label newly synthesized RNAs. As a negative control, one well was pretreated with the rRNA synthesis inhibitor CX-5461 (100 nM, 30 min). Cells were then washed with PBS, and fixed in fixation buffer (4% formaldehyde in PBS) for 15 min. The fixed cells were then permeabilized with permeabilization buffer (0.5% Triton-X 100 in PBS) for 10 min, before three further washes with PBS. Cells were then incubated in the blocking buffer

(4% BSA in PBS, supplemented with SUPERase-In RNase inhibitor) for 30 min, before incubation for 1 h at room temperature with a primary antibody solution of mouse anti-BrdU (1:200) (which also detects FUrd) along with a rabbit anti-nucleolin (1:100), diluted in blocking buffer. This was followed by three PBS washes and incubation with fluorophore-conjugated secondary antibodies (diluted in blocking buffer), along with Hoechst 33258 (1:2,000) for a further 1 h at room temperature in the dark. Slides were then washed three times with PBS and imaged on a Zeiss LSM 880 confocal microscope, using a ×63 oil immersion lens. The microscope was operated using the Zeiss ZEN software (v.3.5). Nucleolar FUrd density levels were quantified from individual nucleoli using ImageJ (v.1.53t), as described previously[61].

### RNA library preparation, sequencing and analysis
Libraries for RNA sequencing were prepared from 700 ng of purified RNA from RNase H-treated or untreated TREX samples, using a Lexogen RiboCop rRNA Depletion HMR V2 kit for ribosomal RNA depletion coupled to a CORALL Total RNA-seq V2 kit. Lexogen UDI 12 nt was used as the indexing system, and the PCR Add-on Kit from Illumina was used to optimize the number of PCR cycles. The PCR-amplified indexed libraries were sequenced to a depth of 40 million reads per sample, using 150 bp paired-end sequencing on an Illumina NovaSeq 6000 instrument (Novogene). The quality of generated FASTQ files was checked with fastqc tool (v.0.11.9). Unique molecular identifiers (UMIs) were extracted from the initial FASTQ files using UMI-tools (v.1.1.1)[74]. This involves the identification and removal of UMIs from each read, allowing for accurate quantification of gene expression and mitigation of amplification biases caused by PCR duplicates. The reads were subjected to trimming with trim-galore (v.0.6.5) to remove adapter sequences and low-quality reads with the following parameters: --paired --retain_unpaired --illumina --gzip. Reads were aligned to the reference human genome GRCh38.p13 using the STAR aligner (v.2.7.9a)[75]. The alignment parameters were set as --outFilterType BySJout --outFilterMultimapNmax 20 --alignSJoverhangMin 8 --alignSJDBoverhangMin 1 --peOverlapNbasesMin 40 --peOverlapMMp 0.8 --outFilterMismatchNoverLmax 0.6 --alignIntronMin 20 --alignIntronMax 1000000 --alignMatesGapMax 1000000 --outSAMattributes NH HI NM MD --outSAMtype BAM SortedBy-Coordinate --quantMode TranscriptomeSAM --twopassMode Basic --outFilterScoreMinOverLread 0 --outFilterMatchNminOverLread 0 --outFilterMismatchNmax 2 --limitOutSJcollapsed 2000000. The resulting alignment files were used to generate a matrix with counts and transcript per million values using the rsem-calculate-expression function from the RSEM software (v.1.3.1)[76], using the following options: --bam --strandedness forward --no-bam-output --paired-end. The transcript per million values were then $\log_2$ transformed and filtered based on having valid values in at least three biological replicates in at least one sample group (RNase H-treated or control), before imputation of missing values using the Perseus Imputation function (downshift, 1.8 s.d.; variation, 0.3 s.d.). The values in each sample group were then averaged and plotted against each other as a scatter plot. The Perseus multidimensional outlier test, using a 0.95 quantile cut-off with Benjamini–Hochberg $P$ value correction, was applied to reveal the significant outliers.

### eCLIP data analysis
Raw eCLIP fastq sequencing datasets for the available proteins of interest were downloaded from the ENCODE Portal[60,77], using the following accessions: ENCFF460RIG, ENCFF826EHK, ENCFF120HCK, ENCFF354DZK, ENCFF912JVC, ENCFF038QQM, ENCFF589NCU, ENCFF361CSS, ENCFF463PBP, ENCFF120UIK, ENCFF974GXQ, ENCFF-498BYQ, ENCFF678KRL, ENCFF813QRH, ENCFF900HNJ, ENCFF-537BAJ, ENCFF161CIM, ENCFF656UJI, ENCFF091OGJ, ENCFF200IHY, ENCFF134DMB, ENCFF044HRW, ENCFF520MEC, ENCFF372THP, ENCFF634ATL, ENCFF317QOW, ENCFF057SJT, ENCFF613QDB,

ENCFF321BPE, ENCFF747FBU, ENCFF692APB, ENCFF595TEO and ENCFF251NBU. These accessions correspond to immunoprecipitation samples and size-matched inputs (SMIs) for SRSF9, GTF2F1 and NOLC1, from HepG2 and K562 cells. Data were processed following the ENCODE eCLIP standard operating procedure (eCLIP-seq Processing Pipeline v.2.2, April 9, 2020), with some modifications, to obtain the protein binding sites on preribosomal 45S RNA. First, UMIs were extracted and appended to the fastq header using UMI-tools (v.1.1.4)[74]. Next, adapters were trimmed twice using Cutadapt (v.4.4) to account for double ligation events, and reads shorter than 18 bp were discarded. Quality of the data was checked after each trimming using FastQC (v.0.12.1). Reads were then mapped to a custom human pre-ribosomal RNA 45S N1 genome file (NCBI Reference Sequence: NR_145819.1) using STAR (v.2.7.10b)[75]. The obtained aligned reads in BAM format were sorted and indexed using SAMtools (v.1.18)[78]. PCR duplicates were removed with UMI-tools based on the presence of identical UMIs on reads mapped to the same position. For paired-end data, only the second read of each pair was selected using SAMtools. Finally, uniquely mapped reads were indexed, and each immunoprecipitation sample BAM file was normalized to the corresponding SMI using deepTools bamCompare (v.3.5.2)[79]. This step accounted for sequencing depth using RPKM normalization, and compared the BAM files directly to output the $\log_2$ ratio of the signals with bin size set to 1 bp. The resulting bedGraph files were displayed in IGV-Web (v.1.13.9)[80] to identify the binding sites enriched over the SMI, with the scale representing the positive fold change.

### Statistical analysis
All proteomics and RNA sequencing statistical analyses were performed by Perseus[73] (v.1.6.2.3), as described above. All other statistical analyses were performed using GraphPad PRISM (v.9.5.1). Unpaired one-way (for CLIP) or two-way (for pre-rRNA steady-state level analyses) Student's $t$-test was applied to all RT–qPCR comparisons. Kruskal–Wallis one-way analysis of variance with Dunn's multiple comparisons test was used for the nascent rRNA imaging analysis.

### Reporting summary
Further information on research design is available in the Nature Portfolio Reporting Summary linked to this article.

## Data availability
All MS raw files and their associated MaxQuant output files were deposited on ProteomeXchange Consortium[81] via the PRIDE partner repository, under the accession numbers PXD044643, PXD045385 and PXD044659 (https://proteomecentral.proteomexchange.org/). All RNA sequencing FASTQ raw files were deposited to the NCBI Bio-Project portal, under the Project accession number PRJNA994065. The Genome Reference Consortium Human Build 38 patch release 13 (GRCh38.p13) database was acquired from https://www.ncbi.nlm.nih.gov/datasets/genome/GCF_000001405.39/. The Uniprot human reference proteome (UP000005640) database was acquired from https://www.uniprot.org/proteomes/UP000005640. Source data are provided with this paper.

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

## Acknowledgements

We thank J. Ule, A. Trupej, K. Kranc, K. Rouault-Pierre, S. Godinho, S. McClelland and T. Sharp for critical comments on the manuscript. We also acknowledge the BCI MS core facility for their support with the proteomics experiments. This work was supported by Medical Research Council grants (MR/P009417/1 and MR/W001500/1) and Barts Charity grants (MGU0346 and G-002420) to F.K.M. L.S. is supported by Barts Charity (MGU0404), Cancer Research UK Career Establishment Award (RCCFEL\100007), Royal Society Research Grant (RGS\R1\231139) and the Academy of Medical Science Springboard Award (SBF006\1026). G.G. is supported by an AIRC Fellowship for Abroad.

## Author contributions

F.K.M. conceived the study. M. Dodel optimized the methodology. M. Dodel, G.G., S.K., Z.R. and E.L.A. performed the experiments. G.G., M. Dermit, E.L.A., F.C., L.S. and F.K.M. analyzed the data. F.K.M. and L.S. acquired funding, supervised the work and wrote the paper.

## Competing interests

F.K.M., L.S., M. Dermit, G.G. and M. Dodel are inventors and contributors on a pending patent covering the TREX method. The other authors declare no competing interests.

## Additional information

**Extended data** is available for this paper at https://doi.org/10.1038/s41592-024-02181-1.

**Correspondence and requests for materials** should be addressed to Lovorka Stojic or Faraz K. Mardakheh.

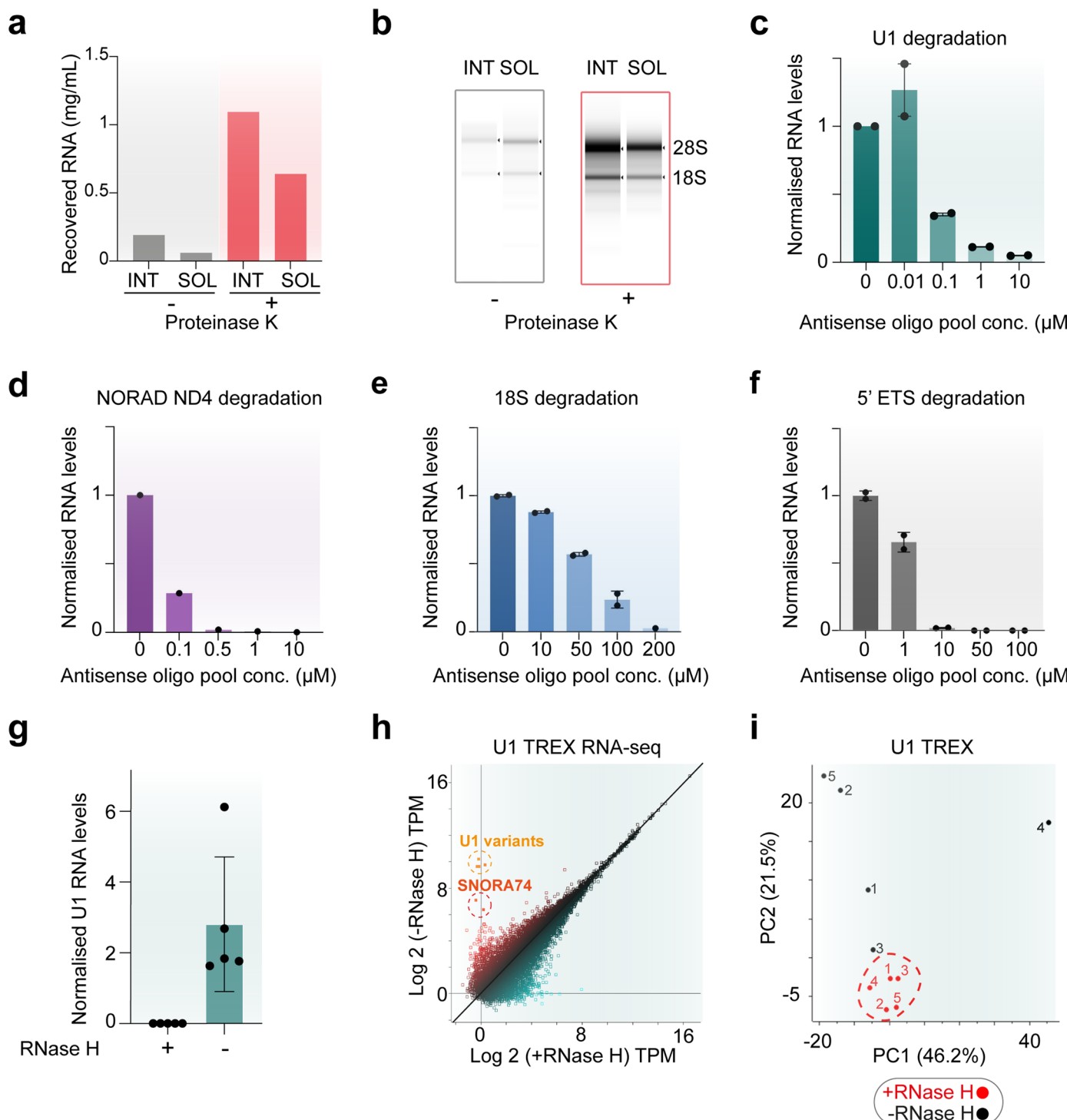

**Extended Data Fig. 1 | See next page for caption.**

**Extended Data Fig. 1 | Validation of TREX parameters. (a)** Confirmation of the solubilization of intact RNA-protein adducts from the interface of crosslinked HCT116 cells. UV-C crosslinked HCT116 cells were lysed in TRIZOL and subjected to organic phase separation to isolate RNA-protein adducts in the interface (INT). The Interface pellets were subsequently solubilized according to the TREX protocol (SOL), before treatment with or without proteinase-K to remove the crosslinked proteins. The mixtures were then subjected to standard TRIZOL-based RNA extraction. Equal volumes from equivalent starting amounts for each INT or SOL sample, with or without proteinase K treatment, were quantified for RNA content using Nanodrop. Interface primarily contains protein-bound RNA, since the majority of RNA is only recoverable after proteinase K digestion. More than half of this protein-bound RNA is solubilized by the TREX resolubilization protocol. **(b)** Analysis of the integrity of the recovered RNA from (a) by capillary electrophoresis. Equal volumes from equal starting amounts for each INT or SOL sample, with or without proteinase K treatment, were resolved by capillary electrophoresis using Tapestation. The solubilized protein-bound RNAs are largely intact, as judged by visualization of full-length 28S and 18S rRNA bands. **(c)** Assessment of the RNase H-mediated degradation of U1 snRNA in solubilized interface fractions, by dose-dependent addition of tiling antisense DNA oligonucleotides. Solubilized protein-bound RNAs from the interface of phase separated UV-C-treated HCT116 cells were heat denatured and annealed to increasing concentrations of a pool of tiling DNA oligonucleotides complementary to the U1 sequence. The annealed interface samples were treated with RNase H to degrade the hybridized RNAs. The remaining amount of U1 snRNA in each sample was then quantified by RT-qPCR, using specific probes against U1. As input control, probes against RPS18 and GAPDH were used, and the U1 levels relative to the two controls were determined using the ΔCt method. Data are presented as mean −/+ range (n = 2 biological replicates). **(d)** Assessment of the RNase H-mediated degradation of NORAD lncRNA in solubilized interface fractions, by dose-dependent addition of tiling antisense DNA oligonucleotides. Solubilized protein-bound RNAs from the interface of phase separated UV-C-treated HCT116 cells were heat denatured and annealed to increasing concentrations of a pool of tiling DNA oligonucleotides complementary to the NORAD ND4 segment. The annealed interface samples were then treated with RNase H to degrade the hybridized RNAs. The remaining amount of NORAD ND4 segment in each sample was quantified by RT-qPCR, using specific probes against this section of NORAD. As an internal control, probes against the 5′ end of NORAD were used, and the NORAD ND4 levels relative to its 5′ end region were determined using the ΔCt method. **(e)** Assessment of the RNase H-mediated

degradation of 18S rRNA in solubilized interface fractions, by dose-dependent addition of tiling antisense DNA oligonucleotides. Solubilized protein-bound RNAs from the interface of phase separated UV-C-treated HCT116 cells were heat denatured and annealed to increasing concentrations of a pool of tiling DNA oligonucleotides complementary to the 18S sequence. The annealed interface samples were then treated with RNase H to degrade the hybridized RNAs. The amount of 18S in each sample was quantified by RT-qPCR, using specific PCR probes against 18S. As an internal control, probes against 28S rRNA were used, and 18S levels relative to the 28S were determined using the ΔCt method. Data are presented as mean -/+ range (n = 2 biological replicates). **(f)** Assessment of the RNase H-mediated degradation of 5′ETS region of pre-rRNA in solubilized interface fractions, by dose-dependent addition of tiling antisense DNA oligonucleotides. Solubilized protein-bound RNAs from the interface of phase separated UV-C-treated HCT116 cells were heat denatured and annealed to increasing concentrations of a pool of tiling DNA oligonucleotides complementary to the 5′ETS sequence. The annealed interface samples were then treated with RNase H to degrade the hybridized RNAs. The amount of 5′ETS pre-rRNA in each sample was quantified by RT-qPCR, using specific PCR probes against 5′ETS. As an internal control, probes against 18S rRNA were used, and 5′ETS levels relative to the 18S were determined using the ΔCt method. Data are presented as mean −/+ range (n = 2 biological replicates). **(g)** Analysis of depletion efficiency in the RNase H-treated vs. untreated samples of the U1 TREX experiment (related to Fig. 1c). Total RNA was extracted from aliquots of RNase H treated and untreated TREX samples, and was subjected to RT-qPCR analysis using specific probes against U1. As input control, probes against RPS18 and GAPDH were used, and U1 levels relative to the two controls were determined in each sample using the ΔCt method. RNase H treatment results in near complete U1 removal. Data are presented as mean −/+ SD (n = 5 biological replicates). **(h)** Analysis of depletion specificity in the RNase H-treated vs. untreated samples of the U1 TREX experiment (related to Fig. 1c). Total extracted RNA from (g) was analyzed by whole-transcriptome RNA-seq to reveal differences following RNase H treatment. Median TPM values for RNase H treated and untreated samples were plotted against each other, with significant outliers identified using the multi-dimensional outlier test from Perseus. Several transcript variants of U1, followed by SNORA74A and B, were the only significantly degraded transcripts detected. **(i)** Principal component analysis (PCA) of the LFQ values from the proteomics analysis of RNase H treated (red) and untreated (black) U1 TREX samples. Five biological replicates per condition were analyzed.

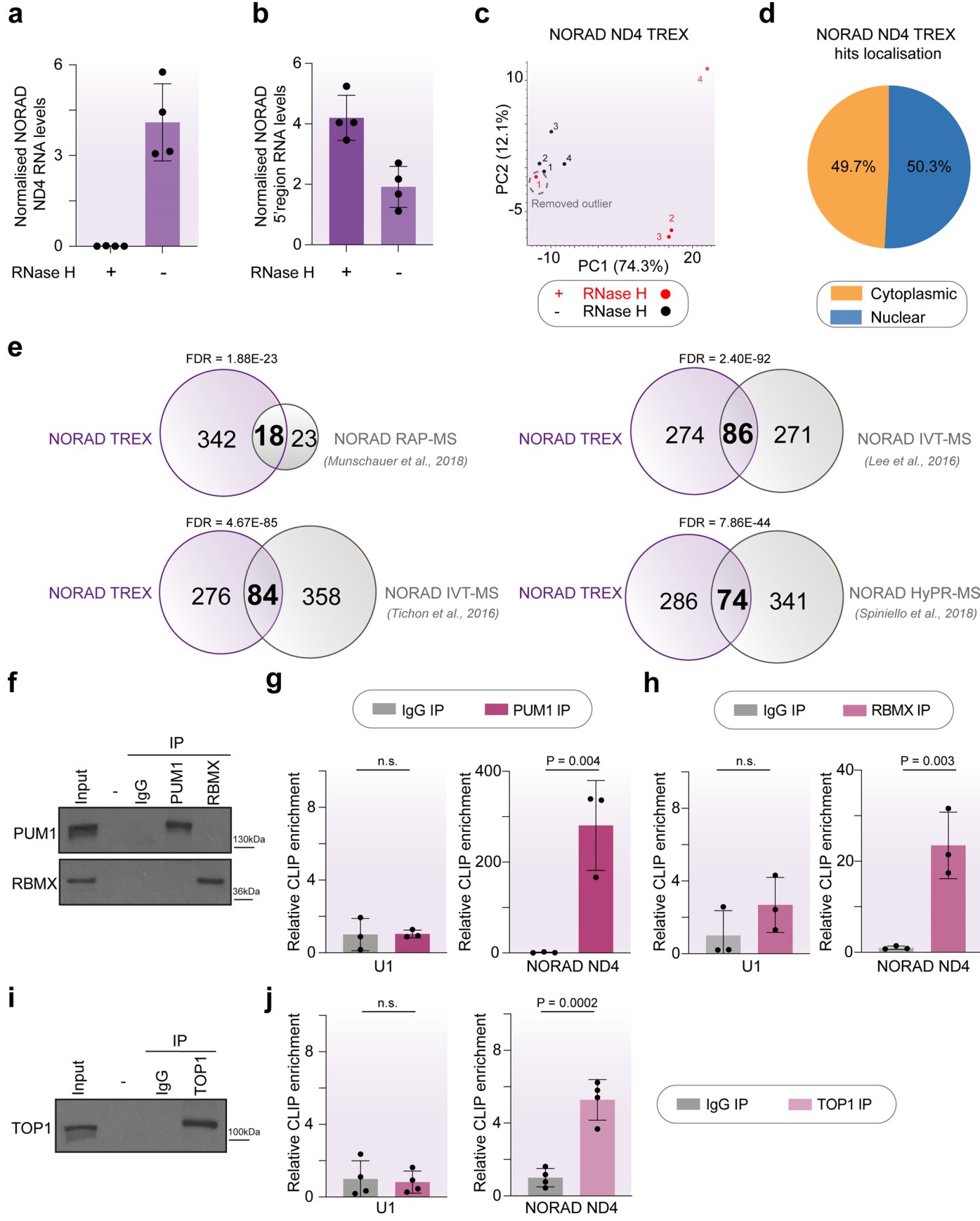

**Extended Data Fig. 2 | See next page for caption.**

**Extended Data Fig. 2 | Analysis of NORAD by TREX. (a)** Analysis of depletion efficiency in the RNase H-treated vs. untreated samples of the NORAD ND4 TREX experiment (related to Fig. 2b). Total RNA was extracted from aliquots of RNase H treated and untreated TREX samples, and was subjected to RT-qPCR analysis using specific probes against the NORAD ND4 domain. As input control, probes against RPS18 and GAPDH were used, and NORAD ND4 levels relative to the two controls were determined in each sample using the ΔCt method. RNase H treatment results in near complete degradation of NORAD ND4 segment. Data are presented as mean −/+ SD (n = 4 biological replicates). **(b)** Analysis of the depletion region-specificity in the NORAD ND4 TREX samples (related to Fig. 2b). The total RNA extracted from NORAD ND4 TREX samples in (a) was subjected to RT-qPCR analysis using specific probes against the 5′ end segment of NORAD. As input control, probes against RPS18 and GAPDH were used, and the levels NORAD 5′ region relative to the two controls were determined in each sample using the ΔCt method. RNAse H treatment does not degrade the 5′ end segment of NORAD. Data are presented as mean −/+ SD (n = 4 biological replicates). **(c)** PCA of the LFQ values from the proteomics analysis of RNase H treated (red) and untreated (black) NORAD ND4 TREX samples. Four biological replicates per condition were analyzed. RNase H-treated sample 1 is an outlier and groups much more closely with the untreated samples, suggesting experiment failure. **(d)** GOCC analysis of the primary subcellular location associated with the significant hits from NORAD ND4 TREX. While 49.7% of the hits were annotated as belonging to the 'cytoplasmic part' category of GOCC, 50.3% were annotated as belonging to the 'nuclear part' category. **(e)** Venn diagram of the overlap between the lists of NORAD ND4 hits identified by TREX, and four previous NORAD interactome capture studies (Lee et al.[28]; Munschauer et al.[11]; Spiniello et al.[32]; Tichon et al.[27]). A highly significant overlap, calculated using Fisher's exact test with

Benjamini-Hochberg FDR estimation, was detected in each comparison. The exact number of overlapping vs. non-overlapping proteins in each comparison, as well as the Fisher's exact test FDR values, are depicted on the Venn diagrams. **(f)** Western blot analysis of HCT116 input lysates, as well as immunoprecipitates (IPs) from non-specific rabbit IgG, PUM1, and RBMX CLIP analyses. Samples were resolved by SDS-PAGE, and analyzed by immunoblotting with the indicated antibodies, showing specific enrichment of each bait protein. The results are representative of 3 independent experiments. **(g)** RT-qPCR analysis of PUM1 and IgG control CLIP samples from HCT116 cells, showing specific enrichment of NORAD ND4 region, but not U1 RNA, in PUM1 IPs. CLIP enrichments are presented as % of input RNA, and normalized relative to IgG control average. Data are presented as mean −/+ SD (n = 3 biological replicates). P-values were calculated using an unpaired one-tailed t-test (n.s.: not significant). **(h)** RT-qPCR analysis of RBMX and IgG control CLIP samples from HCT116 cells, showing specific enrichment of NORAD ND4 region, but not U1 RNA, in RBMX IPs. CLIP enrichments are presented as % of input RNA, and normalized relative to IgG control average. Data are presented as mean −/+ SD (n = 3 biological replicates). P-values were calculated using an unpaired one-tailed t-test (n.s.: not significant). **(i)** Western blot analysis of HCT116 input lysates, as well as IPs from non-specific rabbit IgG and TOP1 CLIP analyses. Samples were resolved by SDS-PAGE, and analyzed by immunoblotting with the indicated antibodies, showing specific enrichment of TOP1. The results are representative of 4 independent experiments. **(j)** RT-qPCR analysis of TOP1 and IgG control CLIP samples from HCT116 cells, showing specific enrichment of NORAD ND4 region, but not U1 RNA, in TOP1 IPs. CLIP enrichments are presented as % of input RNA, and normalized relative to IgG control average. Data are presented as mean −/+ SD (n = 4 biological replicates). P-values were calculated using an unpaired one-tailed t-test (n.s.: not significant).

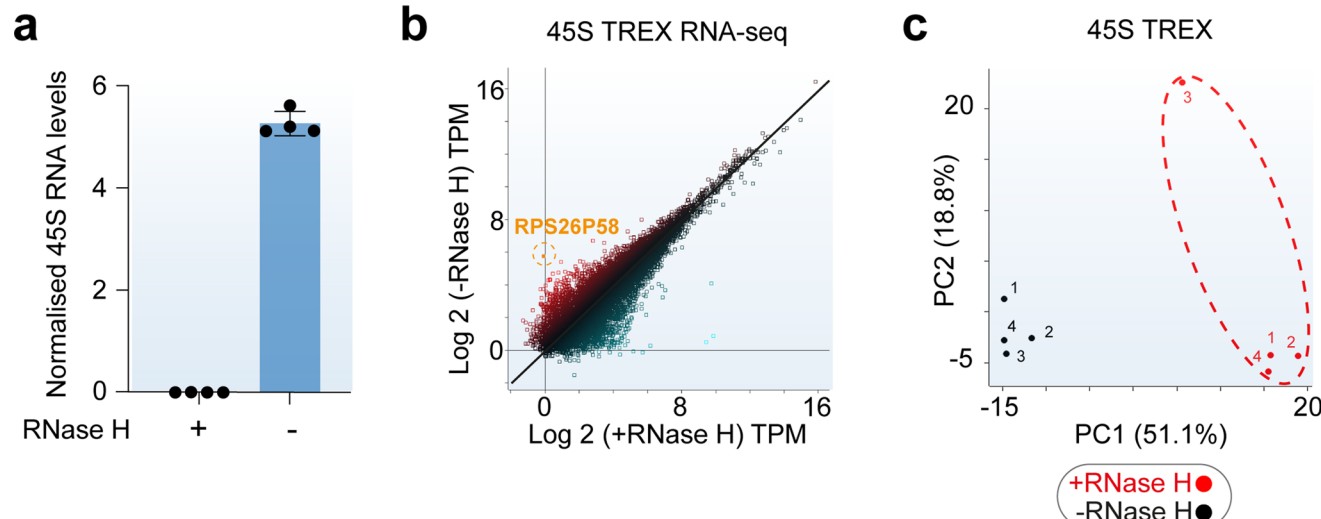

**Extended Data Fig. 3 | Analysis of 45S by TREX. (a)** Analysis of depletion efficiency in the RNase H-treated vs. untreated samples of the 45S rRNA TREX experiment (related to Fig. 3b). Total RNA was extracted from aliquots of RNase H treated and untreated TREX samples, and was subjected to RT-qPCR analysis using specific probes against two segments of 45S rRNA. As input control, probes against RPS18 and GAPDH were used, and 45S rRNA levels relative to the two controls were determined in each sample using the ΔCt method. RNase H treatment results in near complete degradation of 45S rRNA. Data are presented as mean −/+ SD (n = 4 biological replicates). **(b)** Analysis of depletion specificity in the RNase H-treated vs. untreated samples of the 45S TREX experiment (related

to Fig. 3b). Total extracted RNA from (a) was analyzed by whole-transcriptome RNA-seq to reveal differences in the transcriptome following RNase H treatment. Median TPM values for RNase H treated and untreated samples were plotted against each other, with significant outliers identified using the multi-dimensional outlier test from Perseus. The only significant RNase H-degraded transcript detected was that of *RPS26P58* pseudogene (Note that 45S rRNA itself is not detectable in this assay due to the ribo-depletion procedure used in library preparation). **(c)** PCA of the LFQ values from the proteomics analysis of RNase H treated (red) and untreated (black) 45S rRNA TREX samples. Four biological replicates per condition were analyzed.

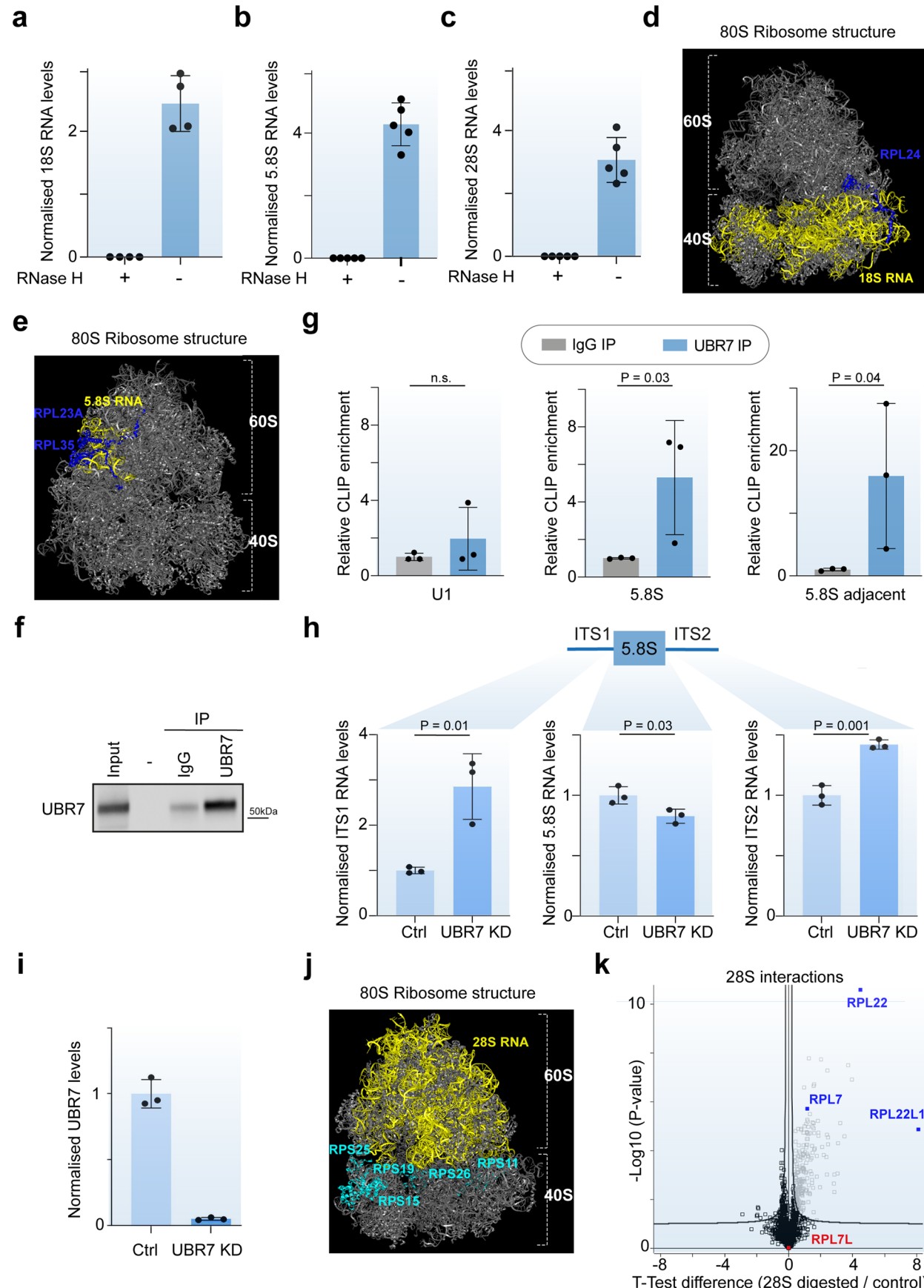

**Extended Data Fig. 4 | See next page for caption.**

**Extended Data Fig. 4 | Analysis of 18S, 5.8S, and 28S rRNA interactomes by TREX. (a)** Analysis of depletion efficiency in the RNase H-treated vs. untreated samples of the 18S rRNA TREX experiment (related to Fig. 4b). Total RNA was extracted from aliquots of RNase H treated and untreated TREX samples, and was subjected to RT-qPCR analysis using specific probes against the 18S segment of 45S rRNA. As input control, probes against RPS18 and GAPDH were used, and 18S rRNA levels relative to the two controls were determined in each sample using the ΔCt method. RNase H treatment results in near complete degradation of 18S rRNA. Data are presented as mean −/+ SD (n = 4 biological replicates). **(b)** Analysis of depletion efficiency in the RNase H-treated vs. untreated samples of the 5.8S rRNA TREX experiment (related to Fig. 4d). Total RNA was extracted from aliquots of RNase H treated and untreated TREX samples, and was subjected to RT-qPCR analysis using specific probes against the 5.8S segment of 45S rRNA. As control, probes against the 18S segment and GAPDH were used, and the relative 5.8S rRNA levels were determined in each sample using the ΔCt method. RNase H treatment results in near complete degradation of the 5.8S segment. Data are presented as mean −/+ SD (n = 5 biological replicates). **(c)** Analysis of depletion efficiency in the RNase H-treated vs. untreated samples of the 28S rRNA TREX experiment (related to Fig. 4e). Total RNA was extracted from aliquots of RNase H treated and untreated TREX samples, and was subjected to RT-qPCR analysis using specific probes against the 28S segment of 45S rRNA. As control, probes against the 18S segment and GAPDH were used, and the relative 28S rRNA levels were determined in each sample using the ΔCt method. RNase H treatment results in near complete degradation of the 28S segment. Data are presented as mean −/+ SD (n = 5 biological replicates). **(d)** Structure of the Human 80S ribosome (*PDB ID:* 4UG0), visualized on PyMOL (version 2.5.4), with the 18S rRNA (yellow) and RPL24 (blue) molecules highlighted on the structure. RPL24 extends from the 60S subunit into the 40S subunit and makes extensive contacts with 18S rRNA. **(e)** Structure of the Human 80S ribosome (*PDB ID:* 4UG0), visualized on PyMOL (version 2.5.4), with the 5.8S rRNA (yellow), RPL23A (blue), and RPL35 (blue) molecules highlighted on the structure. Both RPL23A and RPL35 make extensive direct contacts with the 5.8S rRNA in the 60S subunit. **(f)** Western blot analysis of HCT116 input lysates, as well as immunoprecipitates (IPs) from non-specific rabbit IgG and UBR7 CLIP analyses. Samples were resolved by SDS-PAGE, and analyzed by immunoblotting with the indicated antibodies, showing

specific enrichment of UBR7. The results are representative of 3 independent experiments. **(g)** RT-qPCR analysis of UBR7 and IgG control CLIP samples from HCT116 cells, showing specific enrichment of 5.8S rRNA, but not U1 RNA, in UBR7 IPs. A specific enrichment was also detectable with probes against the last 105 nucleotides of ITS1 that is immediately adjacent to the 5.8S region (5.8S adjacent), suggestive of UBR7 interaction with the premature 5.8S transcript. CLIP enrichments are presented as % of input RNA, and normalized relative to IgG control average. Data are presented as mean −/+ SD (n = 3 biological replicates). P-values were calculated using an unpaired one-tailed t-test (n.s.: not significant). **(h)** RT-qPCR analysis of ITS1, 5.8S, and ITS2 RNA levels in control vs. UBR7 depleted HCT116 cells. HCT116 cells were transfected with non-targeting control (Ctrl) or UBR7 siRNA pools (UBR7 KD) for 72 hrs, followed by total RNA extraction and RT-qPCR analysis using specific probes against the indicated rRNA segments. As control, probes against the RPS18 and GAPDH were used, and the relative RNA levels were determined using the ΔCt method. UBR7 knockdown results in a significant accumulation of ITS1 and ITS2, with concomitant reduction in 5.8S levels, indicative of a defect in 5.8S processing. Data are presented as mean −/+ SD (n = 3 biological replicates). P-values were calculated using an unpaired two-tailed t-test. **(i)** RT-qPCR analysis of UBR7 mRNA levels in control vs. UBR7 depleted HCT116 cells from (h). Total extracted RNA from (h) was subjected to RT-qPCR analysis using specific probes against UBR7 mRNA. As control, probes against the RPS18 and GAPDH were used, and the relative RNA levels were determined in each sample using the ΔCt method, confirming efficient siRNA-mediated knockdown of UBR7. Data are presented as mean −/+ SD (n = 3 biological replicates). **(j)** Structure of the Human 80S ribosome (*PDB ID:* 4UG0), visualized on PyMOL (version 2.5.4), with the 28S rRNA (yellow), RPS11 (cyan), RPS15 (cyan), RPS19 (cyan), RPS25 (cyan), and RPS26 (cyan) highlighted on the structure. All highlighted proteins are located close to the interface of the two ribosomal subunits. **(k)** Volcano plot of the two-sided two-sample t-test comparison of 28S rRNA digested vs undigested TREX samples (n = 5 biological replicates), with RPL22, RPL22L1, RPL7, and RPL7L highlighted on the graph. Curved lines mark the significance boundary (FDR = 0.05, S0 = 0.1). Both RPL22 and RPL22L1 paralogues are detected amongst the significant 28S rRNA interactors, while only RPL7 is detected as an interactor.

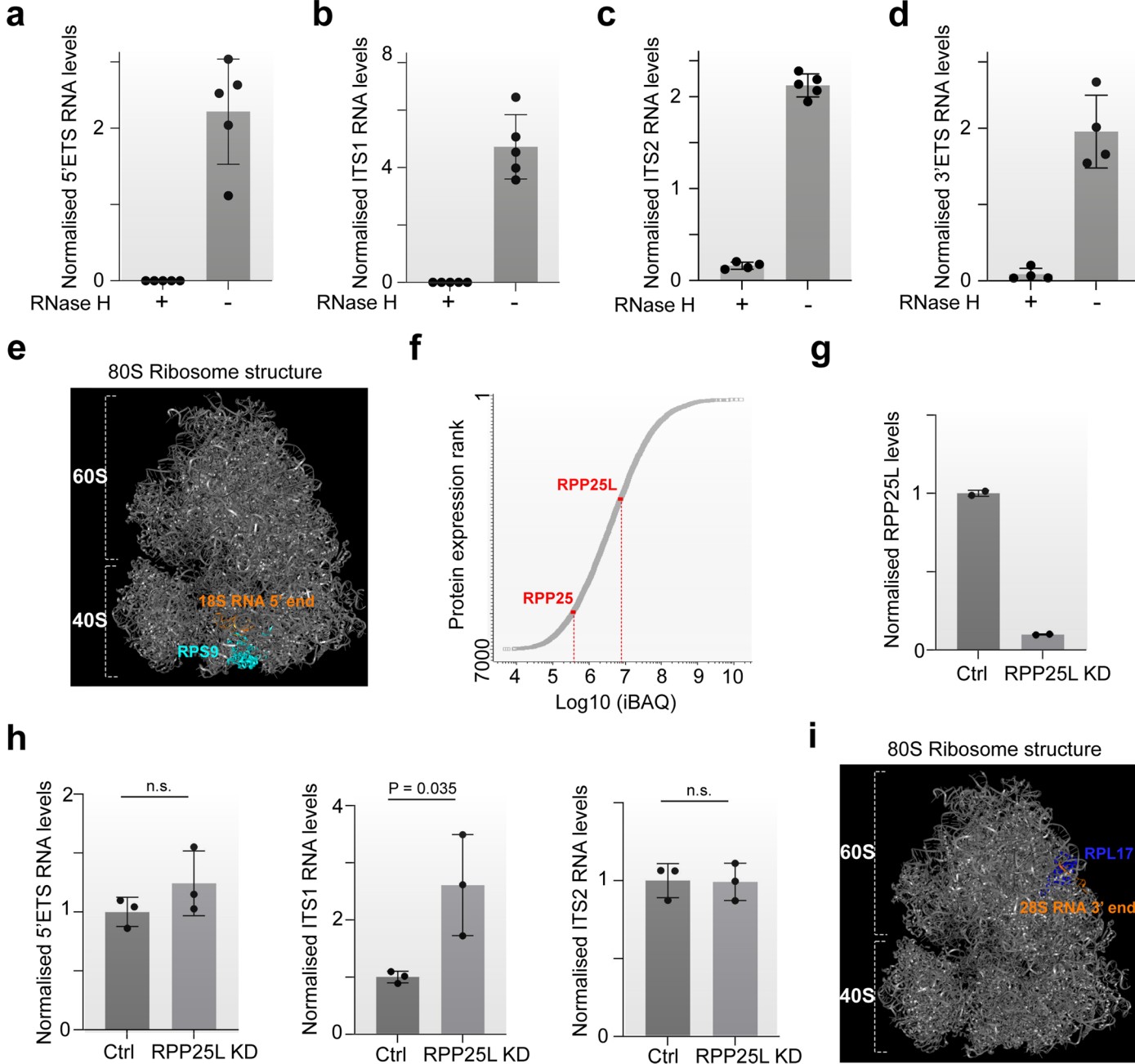

**Extended Data Fig. 5 | See next page for caption.**

**Extended Data Fig. 5 | Analysis of 5'ETS, ITS1, ITS2, and 3'ETS interactomes by TREX. (a)** Analysis of depletion efficiency in the RNase H-treated vs. untreated samples of the 5'ETS rRNA TREX experiment (related to Fig. 5b). Total RNA was extracted from aliquots of RNase H treated and untreated TREX samples, and was subjected to RT-qPCR analysis using specific probes against the 5'ETS segment of 45S rRNA. As control, probes against the 18S segment and GAPDH were used, and the relative 5'ETS rRNA levels were determined in each sample using the ΔCt method. RNAse H treatment results in near complete degradation of the 5'ETS segment. Data are presented as mean −/+ SD (n = 5 biological replicates). **(b)** Analysis of depletion efficiency in the RNase H-treated vs. untreated samples of the ITS1 rRNA TREX experiment (related to Fig. 5d). Total RNA was extracted from aliquots of RNase H treated and untreated TREX samples, and was subjected to RT-qPCR analysis using specific probes against the ITS1 segment of 45S rRNA. As control, probes against the 18S segment and GAPDH were used, and the relative ITS1 rRNA levels were determined in each sample using the ΔCt method. RNAse H treatment results in near complete degradation of the ITS1 segment. Data are presented as mean −/+ SD (n = 5 biological replicates). **(c)** Analysis of depletion efficiency in the RNase H-treated vs. untreated samples of the ITS2 rRNA TREX experiment (related to Fig. 5f). Total RNA was extracted from aliquots of RNase H treated and untreated TREX samples, and was subjected to RT-qPCR analysis using specific probes against the ITS2 segment of 45S rRNA. As control, probes against the 18S segment and GAPDH were used, and the relative ITS2 rRNA levels were determined in each sample using the ΔCt method. RNAse H treatment results in near complete degradation of the ITS2 segment. Data are presented as mean −/+ SD (n = 5 biological replicates). **(d)** Analysis of depletion efficiency in the RNase H-treated vs. untreated samples of the 3'ETS rRNA TREX experiment (related to Fig. 5h). Total RNA was extracted from aliquots of RNase H treated and untreated TREX samples, and was subjected to RT-qPCR analysis using specific probes against the 3'ETS segment of 45S rRNA. As control, probes against the 18S segment and GAPDH were used, and the relative 3'ETS rRNA levels were determined in each sample using the ΔCt method. RNAse H treatment results in near complete degradation of the 3'ETS segment. Data are presented as mean −/+ SD (n = 4 biological replicates). **(e)** Structure of the Human 80S ribosome (*PDB ID:* 4UG0), visualized on PyMOL (version 2.5.4), with the 5' end of 18S rRNA (orange) and RPS9 (cyan) highlighted on the structure. RPS9 binds specifically at the 5'end of 18S. **(f)** The ranked plot of the iBAQ absolute protein abundance measurements from the total proteome of HCT116, revealing RPP25L to be > 20 fold more expressed than RPP25. **(g)** RT-qPCR analysis of RPP25L mRNA levels in control vs. RPP25L depleted HCT116 cells. HCT116 cells were transfected with non-targeting control (Ctrl) or RPP25L siRNA pools (RPP25L KD) for 72 hrs, followed by total RNA extraction and RT-qPCR analysis using specific probes against RPP25L mRNA. As control, probes against the RPS18 and GAPDH were used, and the relative RNA levels were determined in each sample using the ΔCt method. Data are presented as mean -/+ range (n = 2 biological replicates). **(h)** RT-qPCR analysis of 5'ETS, ITS1, and ITS2 spacer RNA levels in control vs. RPP25L depleted HCT116 cells from (g). Total RNA extracts from were subjected to RT-qPCR analysis using specific probes against the indicated rRNA segments. As control, probes against the RPS18 and GAPDH were used, and the relative RNA levels were determined in each sample using the ΔCt method. RPP25L knockdown only results in a significant accumulation of ITS1. Data are presented as mean -/+ SD (n = 3 biological replicates). P-values were calculated using an unpaired two-tailed t-test (n.s.: not significant). **(i)** Structure of the Human 80S ribosome (*PDB ID:* 4UG0), visualized on PyMOL (version 2.5.4), with the 3' end of 28S rRNA (orange) and RPL17 (blue) highlighted on the structure. RPL17 binds specifically at the 3'end of 28S.

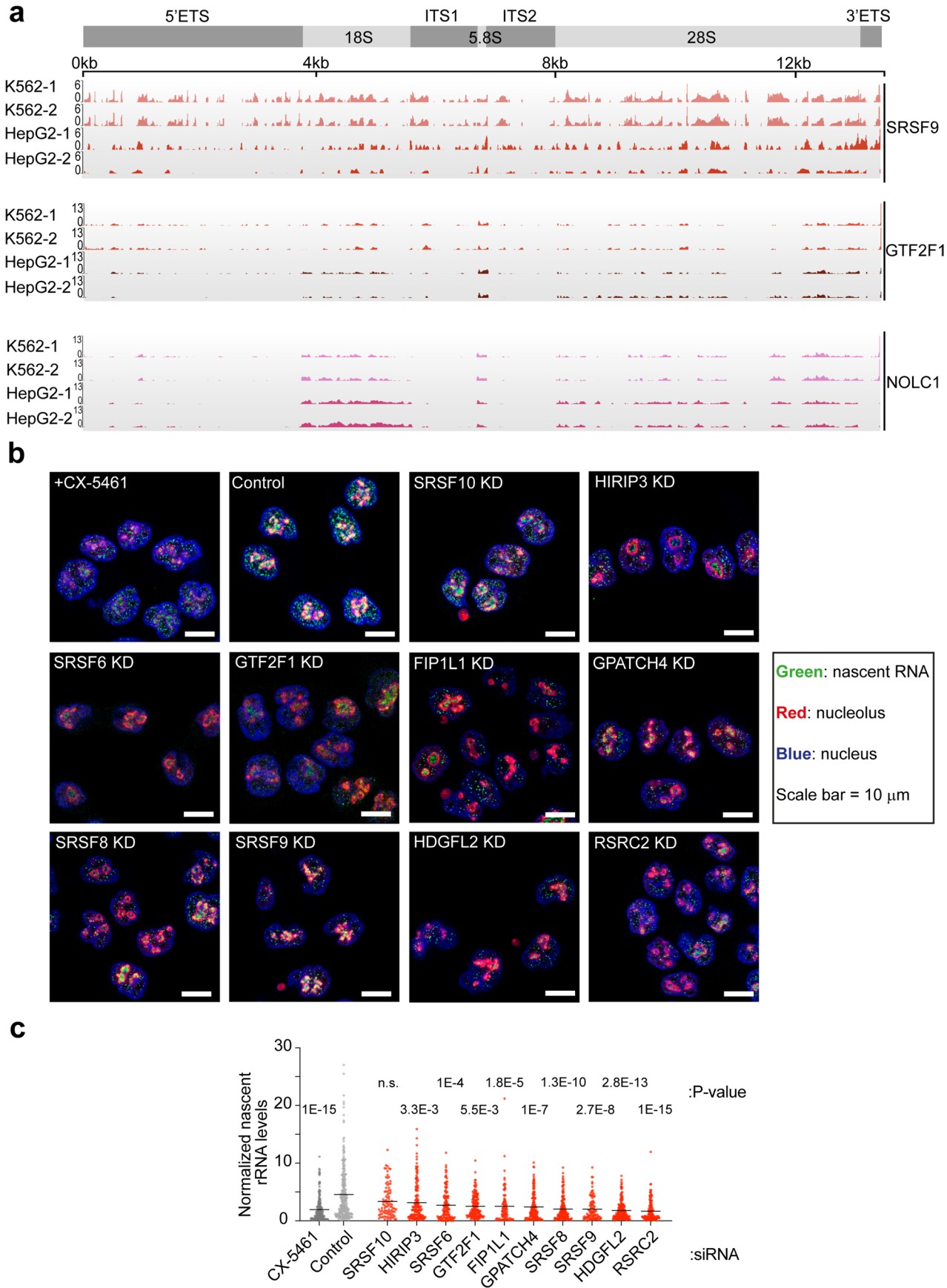

**Extended Data Fig. 6 | See next page for caption.**

**Extended Data Fig. 6 | Analysis of the role of cluster 9 proteins in rRNA regulation. (a)** eCLIP distribution of SRSF9, GTF2F1, and NOLC1 binding sites, across the annotated human *45S* genomic region. Existing eCLIP datasets and their associated controls from two independent replicate experiments in two independent cell lines (HepG2 and K562), were extracted from ENCODE and processed as described in the 'Experimental procedures', in order to reveal the binding sites of each protein across the full length of 45S rRNA. The scales represent the positive fold change relative to the control samples for each experiment. SRSF9, GTF2F1, and NOLC1 exhibit a promiscuous binding pattern across the full length of 45S rRNA. **(b)** Representative images of nascent RNA imaging analysis of HCT116 cells, transfected with a non-targeting control siRNA pool, or siRNA pools against the indicated target genes. Cells were transfected for 72 hrs with the indicated siRNAs, prior to pulse labeling with FUrd (2mM, 30 min) to label nascent RNAs. As a negative control, CX5461 (100nM, 30 min) was used to block rRNA synthesis, prior to the FUrd pulse. Cells were then fixed and immunostained with an anti-FUrd antibody (green) to visualize nascent RNA, along with an anti-nucleolin antibody to reveal the nucleolar boundaries (red), and Hoechst (blue) as the Nuclear marker. Confocal microscopy analysis revealed that knockdown of most investigated proteins, as well as the CX-5461 treatment, significantly inhibited rRNA synthesis, as indicated by reduced nascent RNA levels in the nucleoli. Scale bar = 10 μm. The cell images are representative of at least 50 individual imaged cells per each condition, taken over 2 independent experiments. **(c)** Quantification of nascent rRNA levels in the nucleoli of HCT116 cells from the experiment shown in (b). The nucleolar FUrd levels, indicative of nascent rRNA, were quantified from individual nucleoli in each treatment condition. Dash lines mark the mean value. 90–263 nucleoli per each condition, were pooled and quantified from 2 independent experiments. P-values were calculated using non-parametric one-way ANOVA with Kruskal-Wallis Multiple comparisons test. (n.s.: not significant).

# Reporting Summary

## Statistics

For all statistical analyses, confirm that the following items are present in the figure legend, table legend, main text, or Methods section.

| n/a | Confirmed | |
|---|---|---|
| ☐ | ☒ | The exact sample size (*n*) for each experimental group/condition, given as a discrete number and unit of measurement |
| ☐ | ☒ | A statement on whether measurements were taken from distinct samples or whether the same sample was measured repeatedly |
| ☐ | ☒ | The statistical test(s) used AND whether they are one- or two-sided *Only common tests should be described solely by name; describe more complex techniques in the Methods section.* |
| ☐ | ☒ | A description of all covariates tested |
| ☐ | ☒ | A description of any assumptions or corrections, such as tests of normality and adjustment for multiple comparisons |
| ☐ | ☒ | A full description of the statistical parameters including central tendency (e.g. means) or other basic estimates (e.g. regression coefficient) AND variation (e.g. standard deviation) or associated estimates of uncertainty (e.g. confidence intervals) |
| ☐ | ☒ | For null hypothesis testing, the test statistic (e.g. *F*, *t*, *r*) with confidence intervals, effect sizes, degrees of freedom and *P* value noted *Give P values as exact values whenever suitable.* |
| ☒ | ☐ | For Bayesian analysis, information on the choice of priors and Markov chain Monte Carlo settings |
| ☒ | ☐ | For hierarchical and complex designs, identification of the appropriate level for tests and full reporting of outcomes |
| ☒ | ☐ | Estimates of effect sizes (e.g. Cohen's *d*, Pearson's *r*), indicating how they were calculated |

*Our web collection on statistics for biologists contains articles on many of the points above.*

## Software and code

Policy information about availability of computer code

| Data collection | Thermo XCalibur (version 4.5) SP1 for mass spectrometry data collection; Zeiss ZEN blue (version 3.5) for microscopy imaging data collection. |
|---|---|
| Data analysis | Maxquant (version 1.6.6.3); Perseus (version 1.6.2.3); DescTools (version 0.99.52); fastqc tool (versions 0.11.9 & 0.12.1); UMItools (versions 1.1.1 & 1.1.4); Trim-galore (version 0.6.5); STAR aligner (versions 2.7.9a and 2.7.10b); RSEM software (version 1.3.1); Cutadapt (version 4.4) ; SAMtools (version 1.18); deepTools bamCompare (version 3.5.2); IGV-Web (version 1.13.9); GraphPad PRISM (version 9.5.1); ImageJ (version 1.53t); PyMOL (version 2.5.4) |

For manuscripts utilizing custom algorithms or software that are central to the research but not yet described in published literature, software must be made available to editors and reviewers. We strongly encourage code deposition in a community repository (e.g. GitHub). See the Nature Portfolio guidelines for submitting code & software for further information.

# Data

Policy information about availability of data

All manuscripts must include a data availability statement. This statement should provide the following information, where applicable:

- Accession codes, unique identifiers, or web links for publicly available datasets
- A description of any restrictions on data availability
- For clinical datasets or third party data, please ensure that the statement adheres to our policy

All mass spectrometry raw files and their associated MaxQuant output files were deposited on ProteomeXchange Consortium, via the PRIDE partner repository, under the accession numbers PXD044643, PXD045385, and PXD044659 (https://proteomecentral.proteomexchange.org/). All RNA-sequencing FASTQ raw files were deposited to the NCBI BioProject portal, under the Project accession number PRJNA994065 (https://www.ncbi.nlm.nih.gov/bioproject/PRJNA994065). The Genome Reference Consortium Human Build 38 patch release 13 (GRCh38.p13) database was acquired from (https://www.ncbi.nlm.nih.gov/datasets/genome/ GCF_000001405.39/). The Uniprot human reference proteome (UP000005640) database was acquired from (https://www.uniprot.org/proteomes/UP000005640).

# Human research participants

Policy information about studies involving human research participants and Sex and Gender in Research.

| Reporting on sex and gender | Not relevant to this study |
|---|---|
| Population characteristics | Not relevant to this study |
| Recruitment | Not relevant to this study |
| Ethics oversight | No ethics approval required for this study |

Note that full information on the approval of the study protocol must also be provided in the manuscript.

# Field-specific reporting

Please select the one below that is the best fit for your research. If you are not sure, read the appropriate sections before making your selection.

☒ Life sciences  ☐ Behavioural & social sciences  ☐ Ecological, evolutionary & environmental sciences

For a reference copy of the document with all sections, see nature.com/documents/nr-reporting-summary-flat.pdf

# Life sciences study design

All studies must disclose on these points even when the disclosure is negative.

| Sample size | In this study, pre-determining the sample size was not feasible. However, drawing from standard protocols in protein-interaction experiments like IP-MS, which typically employ three biological replicates, we hypothesized that TREX experiments would be analogous. Consequently, to attain sufficient statistical power, at least three replicate experiments were deemed necessary. To safeguard against potential loss of statistical power due to a failed replicate, we always conducted a minimum of four biological replicates in our TREX experiments. |
|---|---|
| Data exclusions | Amongst all the replicate samples analysed, only one replicate of the NORAD TREX experiment, and one replicate of the ITS2 TREX experiment were excluded, as these clearly failed the sample prep and not much proteins was detected in the resulting data afterwards. Accordingly, PCA analysis clearly showed that the replicates in question had failed and behaved as an outlier (see Extended Data Fig. 2c). No other data was excluded. |
| Replication | The TREX experiments were conducted four to five times each, and statistical analysis encompassed all replicates collectively. Every replication attempt succeeded across all targets, except for one replicate in the NORAD TREX experiment (with three remaining successful) and one in the ITS2 TREX experiment (with four remaining successful). Both failures were attributed to sample loss during preparation, due to lack of protein IDs. These unsuccessful replicates were subsequently excluded from the analyses, as detailed in Extended Data Fig. 2c, and Supplementary Datasets 2 and 9. |
| Randomization | The TREX samples were always randomly allocated for receiving treatment (+RNase H) or no treatment (control) at the digestion step. |
| Blinding | Where possible, the Perseus data analysis step in TREX was carried out blindly by a different researcher (not knowing which sample group is RNase H treated and which is not). Blinding was not implemented in other experiments, as typically the same researcher executed all the stages of an experiment, including analysis. |

# Reporting for specific materials, systems and methods

We require information from authors about some types of materials, experimental systems and methods used in many studies. Here, indicate whether each material, system or method listed is relevant to your study. If you are not sure if a list item applies to your research, read the appropriate section before selecting a response.

## Materials & experimental systems

| n/a | Involved in the study |
|---|---|
| ☐ | ☒ Antibodies |
| ☐ | ☒ Eukaryotic cell lines |
| ☒ | ☐ Palaeontology and archaeology |
| ☒ | ☐ Animals and other organisms |
| ☒ | ☐ Clinical data |
| ☒ | ☐ Dual use research of concern |

## Methods

| n/a | Involved in the study |
|---|---|
| ☒ | ☐ ChIP-seq |
| ☒ | ☐ Flow cytometry |
| ☒ | ☐ MRI-based neuroimaging |

## Antibodies

| | |
|---|---|
| Antibodies used | Please note that the details of all antibodies used in this study, including their dilution, are now provided in Supplementary Table 3, as well as being listed below:<br><br>Anti-Brdu (Merck - B2531-100UL )(1:200 - IF);<br>Anti-Nucleolin (Abcam - ab22758) (1:100 - IF);<br>Alexa Fluor 488-conjugated Donkey Anti-Mouse IgG (H+L) (Jackson ImmunoResearch - 715-545-150) (1:200 - IF);<br>Alexa Fluor 647-conjugated Donkey Anti-Rabbit IgG (H+L) (Jackson ImmunoResearch - 711-605-152) (1:200 - IF);<br>Rabbit IgG non-specific control (Cell Signaling - 2729) (1:200 - CLIP);<br>Topoisomerase1 Antibody (Novus - NBP1-30481) (1:200 - CLIP, 1:2000 -WB);<br>RBMX Antibody (Cell Signaling - 14794) (1:200 - CLIP, 1:1000 - WB);<br>PUM1 Antibody (Proteintech - 26256-1-AP) (1:200 - CLIP, 1:1000 - WB);<br>UBR7 Antibody (Cambridge Biosciences - A304-130A) (1:200 - CLIP, 1:2000 - WB);<br>TrueBlot® Anti-Rabbit IgG HRP (Rockland - 18-8816-31) (1:1000 - WB);<br>Rabbit IgG HRP (GE Healthcare - NA934) (1:5000 - WB); |
| Validation | Anti-Brdu was validated in house and the control experiment is included in the manuscript (CX-5461 treatment, Extended Data Fig. 6b and 6c); Anti-Nucleolin was validated in house in our previous publication (Azman et al., EMBO J, 2023 - Fig.4); Topoisomerase1, PUM1, RBMX, and UBR7 antibodies for CLIP were also validated in-house by western blotting, showing specific immunoprecipitation of their targets at the correct molecular weights (Extended Data Fig. 2f, 2i, and 4f). All the secondary antibodies in use for IF or WB are validated by the manufacturer. |

## Eukaryotic cell lines

Policy information about cell lines and Sex and Gender in Research

| | |
|---|---|
| Cell line source(s) | HCT116 colon carcinoma cells (ATCC - catalogue number: CCL-247) |
| Authentication | STR profiling (latest: 21/07/2023) |
| Mycoplasma contamination | Cells were regulary tested for mycoplasma and were always free of contamination |
| Commonly misidentified lines<br>(See ICLAC register) | None used |

