## [Peer Review File · Nature Methods]

Peer Review Information

Manuscript Title: TREX reveals proteins that bind to specific RNA regions in living cells

Corresponding author name(s): Faraz Mardakheh

Editorial Notes:

Reviewer Comments & Decisions:

Decision Letter, initial version:

27th Sep 2023

Dear Dr Mardakheh,

Your Article, "TREX reveals proteins that bind to specific RNA regions in living cells", has now been seen by 3 reviewers. As you will see from their comments below, although the reviewers find your work of considerable potential interest, they have raised a number of technical concerns. We are interested in the possibility of publishing your paper in Nature Methods, but would like to consider your response to these concerns before we reach a final decision on publication.

We therefore invite you to revise your manuscript to address these concerns, in particularly all technical concerns about TREX's performance should be fully addressed.

[REDACTED]

We hope to receive your revised paper within 12 weeks. If you cannot send it within this time, please let us know. In this event, we will still be happy to reconsider your paper at a later date so long as nothing similar has been accepted for publication at Nature Methods or published elsewhere.

OPEN SCIENCE REQUIREMENTS

REPORTING SUMMARY AND EDITORIAL POLICY CHECKLISTS

Please note that these forms are dynamic ‘smart pdfs’ and must therefore be downloaded and completed in Adobe Reader. We will then flatten them for ease of use by the reviewers. If you would like to reference the guidance text as you complete the template, please access these flattened versions at <http://www.nature.com/authors/policies/availability.html>.

IMAGE INTEGRITY

DATA AVAILABILITY

All novel DNA and RNA sequencing data, protein sequences, genetic polymorphisms, linked genotype and phenotype data, gene expression data, macromolecular structures, and proteomics data must be deposited in a publicly accessible database, and accession codes and associated hyperlinks must be provided in the “Data Availability” section.

Please include a “Data availability” subsection in the Online Methods. This section should inform readers about the availability of the data used to support the conclusions of your study, including accession codes to public repositories, references to source data that may be published alongside the paper, unique identifiers such as URLs to data repository entries, or data set DOIs, and any other statement about data availability. At a minimum, you should include the following statement: “The data that support the findings of this study are available from the corresponding author upon request”, describing which data is available upon request and mentioning any restrictions on availability. If DOIs are provided, please include these in the Reference list (authors, title, publisher (repository name), identifier, year). For more guidance on how to write this section please see: <http://www.nature.com/authors/policies/data/data-availability-statements-data-citations.pdf>

CODE AVAILABILITY

Please include a “Code Availability” subsection in the Online Methods which details how your custom code is made available. Only in rare cases (where code is not central to the main conclusions of the paper) is the statement “available upon request” allowed (and reasons should be specified).

SUPPLEMENTARY PROTOCOL

To help facilitate reproducibility and uptake of your method, we ask you to prepare a step-by-step Supplementary Protocol for the method described in this paper. We  encourage authors to share their step-by-step experimental protocols on a protocol sharing platform of their choice and report the protocol DOI in the reference list. Nature Portfolio's Protocol Exchange is a free-to-use and open resource for protocols; protocols deposited in Protocol Exchange are citable and can be linked from the published article. More details can be found at <https://www.nature.com/protocolexchange/about> www.nature.com/protocolexchange/about.

ORCID

Nature Methods is committed to improving transparency in authorship. As part of our efforts in this direction, we are now requesting that all authors identified as 'corresponding author' on published papers create and link their Open Researcher and Contributor Identifier (ORCID) with their account on the Manuscript Tracking System (MTS), prior to acceptance. This applies to primary research papers only. ORCID helps the scientific community achieve unambiguous attribution of all scholarly contributions. You can create and link your ORCID from the home page of the MTS by clicking on 'Modify my Springer Nature account'. For more information please visit <http://www.springernature.com/orcid>.

Sincerely,
Lei

Lei Tang, Ph.D.
Senior Editor
Nature Methods

Reviewers' Comments:

Reviewer #1:
Remarks to the Author:

The authors describe, TREX, a novel approach for identifying the proteins binding an endogenous RNA of interest within cells. The approach builds on previous methods to isolate crosslinked protein-RNA aggregates and extends it by utilizing RNase-H-mediated release of the proteins associated with a specific region. This is a clever idea and it appears to work very well, at least on relatively abundant RNAs. The approach addresses a very important need in the community for methods that can identify proteins binding to a specific region within a long RNA. This is an important problem as many RBPs are promiscuous and so the set of proteins that bind the entire RNA is often very large and it is very difficult to distill from it the proteins that are functionally relevant for further interrogation. The method thus addresses an urgent and unmet need in the community, and I expect it to become widely used, and so it is a great fit for Nature Methods.

The paper is well written, and the presented results appear to be convincing and give some interesting insights into the biology of both NORAD and rRNAs, giving leads to further studies. Overall, I found the level of detail describing these results to be too high, and the discussion is quite repetitive, mainly going over the many proteins found in each part and referencing other studies, so the manuscript can be shortened quite substantially giving more focus to the new method, and to the particular new insights obtained, and making the text more readable (potentially big chunks of the text can become Supplementary Notes).

I have few specific requests:

- The authors should indicate in the Results section the number of cells used in each TREX experiment (including for NORAD, rRNA etc.). Where applicable, it can also be compared with the number of cells used in previous RAP-MS studies (e.g., for NORAD).
- The authors discuss the possible relationship between TOP1 they identify as binding ND4 in NORAD and RBMX which was reported previously by Munschauer et al., but it is important to mention that RBMX binding site is quite far from ND4, and is rather at the 5' end of NORAD.
- I found the annotation enrichment presentation in Fig. 4C,F, Fig. 5C-I and elsewhere to be quite confusing, and the relative position of the category on the X/Y axis doesn't really mean much. A simpler presentation of the most enriched categories, color-coded by the significance would make things simpler yet easier to comprehend.
- Line 467 - "more than 95% of the interacting proteins (381 out of 400) displayed highly specific positional interaction patterns" – how is the specificity defined here? What does it mean that the binding was "highly specific"? What are the effect sizes and the significance of the specificity?
- Sup figS1 D-G: please add normalization to other parts of the same transcript to show the specificity of degradation by probes.
- Was total lysate (line 690) prepared from crosslinked or non crosslinked sample, please clarify in text.

Reviewer #2:

Remarks to the Author:

This manuscript describes a novel method to characterize the proteins bound to specific regions of a target RNA. It does this by first isolating RNA-RBP complexes from crosslinked cells and then releasing the proteins bound to target regions by incubating with antisense DNA and digesting RNA-DNA hybrids using RNaseH. The robustness of this method is established through experiments profiling snRNAs, lncRNAs, and rRNAs. The analysis of different regions of 45S-rRNA and spacer regions of 45S pre-rRNA highlights the power of this technology and how it breaks through current limitations in identifying RNA-RBP interactions. Overall, the paper contains high-quality data, the results are significant, and the claims are generally supported. However, a few issues need to be addressed before this paper can be accepted.

1. In the authors' description of TREX in lines 91-93, it is not clear that they isolate the RNA-protein adduct interface and resolubilize before adding the tiling DNA antisense molecules. This makes the description of the method a bit confusing. The authors should consider amending this to ensure it is clear to the reader.
2. When introducing U2 snRNA as well as the ribosomal RNA molecules and spacer regions profiled, the authors may want to consider discussing the length of the RNAs and the regions they profile. The authors should consider adding this as it would give a better sense of the capabilities of the technology.
3. Similarly, the authors should discuss the length of the DNA tiles they selected and justify why they selected this length. While this is accessible in the supplement, considering this is a paper focused on showcasing a new method, this would be important to elaborate on.
4. One of the major advantages the authors claim that TREX has over RNA-centric methods profiling RNA-RBP interactions is input amount. The authors specifically claim that TREX was able to achieve superior performance with as low as 5 to 10% of the input required by other methods. I believe this claim is overstated. The authors used their ability to profile U1 snRNA from 10 million cells to support their assertion that they have greater sensitivity than others. They cited three other papers that also profiled U1 but that employed more than 100 million cells for that purpose. However, a huge factor in determining the number of cells that are required for successful analysis of a particular RNA target is its cellular abundance. All three of the papers cited characterizing the U1 snRNA interactome used it as a control experiment because its interactome is well-characterized and the U1 snRNA is highly abundant (> 1 million copies per cell). All three papers then used the same number of cells (>100 million) to target lncRNAs, which are generally much less abundant.

When the authors of this paper went after a lower abundance lncRNA target (NORAD, ~1000 copies per cell), they also required 100 million cells, similar to the three other studies. They were successful using

10 million cells for U1 snRNA because of its high abundance, but similarly to the other studies, needed 100 million cells for low abundance targets. Thus their claim of greater sensitivity is not well-supported. In general, this paper is excellent and the technology to assess sequence specific RNA-RBP interactions is interesting and important, and the authors make a rigorous case for their method. However, I strongly believe that the claims of greater sensitivity are overstated. I would recommend the authors remove this claim as it is not needed to make a case for their work and may in fact weaken the paper by reducing its credibility.

Reviewer #3:

Remarks to the Author:

In the manuscript "TRES reveals proteins that bind to specific RNA regions in living cells", Dodel, Giuducci, and colleagues introduce TRES, a method designed to isolate and identify RNA binding proteins interacting with specific regions of a given RNA molecule of interest. The described method is an extension of a set of related protocols that enrich for crosslinked and denatured ribonucleoprotein complexes at an organic/aqueous interface and allow for the global identification of RNA binding proteins (PTex/OOPS/XRNAX). The authors add a step to this protocol by recovering denatured RNPs from the interface and specifically degrading defined stretches of RNA by hybridizing antisense DNA oligos and treating with RNase H. RBPs that were interacting with the RNA stretch of interest are now free to migrate into an organic phase and can be identified by mass spectrometry. The authors benchmark this elegant approach by identifying RBPs interacting with U1 RNA, a ~500 nt stretch of the well-characterized NORAD lncRNA, as well as the entire 45S rRNA (that contains the 5.8S, 18S, and 28S rRNAs, as well as linker regions). Overall the method appears promising in that it may indeed enrich for RBPs that were interacting with the regions of interest, as they do recover previously known interactors. The approach is straightforward and appears to be reproducible and adaptable for use in most molecular biology laboratories (that have access to mass spectrometry, of course). That said, enthusiasm for the manuscript is dampened by the incomplete characterization of specificity and recovery of noise throughout the manuscript. As their approach will most likely be used to try to identify specific regulators of interesting RNA molecules, careful and critical analysis of the false discovery rates is crucial to demonstrate usefulness.

1. the authors recover most of the known interactors of U1. Considering that U1 is a very abundant RNA, question is why are two of the canonical snRNPs missing? Can they be identified by mass spec in principle (trypsin digestion problems or the like)? The authors do not comment on the proteins that are not snRNPs and are not part of the splicing machinery - why are they enriched with the U1 RNP? Is this a universal background that they also find in the other datasets?

2. the authors analyze a fairly long (~500 nt region) of NORAD and recover hundreds of proteins as potential interactors. Among them are the known interactors (PUM1/2) that the Mendell lab found as well as RBMX/SAM68/IGF2BP - interactors that the Lander found in a conflicting manuscript. It appears clear from the reading of the literature that the interaction with PUM proteins is best characterized, as there were plenty of follow-up studies - also consistent with the mainly cytoplasmic localization of the lncRNA. Why do the authors recover what are mainly nuclear RBPs (e.g. RBMX, and plenty of others in the dataset). Perhaps performing TREX from nuclear and cytoplasmic extracts could clarify whether these interactions just reflect different compartments that NORAD could localize in (and thereby also clarify some contention in the literature). Very concerning is that the volcano plot shows moderate enrichment for PUM1 - why are other proteins so much more enriched? Is that reflective of real binding? Some validation should be performed to convince the reader that we are not dealing with background.

3. In all their experiments, the authors also identify previously unappreciated interactors for the RNAs of interest - even rivaling or exceeding enrichment of known interactors. The authors should validate these interactions - not with RIP, as this recovers secondary interactions and does not isolate the specific binding sites - but with CLIP. CLIP as an orthogonal approach can demonstrate that an identified RBP indeed bound at a sequence element in the RNA region digested by RNase H.

4. to be very useful in the identification of regulatory RBPs - rather than recovering a substantial amount of RBPs, as in the case of NORAD - the authors could use shorter tiles; e.g. in NORAD target one of the PREs and show recovery of PUM1 and a region with no PRE and show that no Pumilio proteins are identified by TREX.

Author Rebuttal to Initial comments

Point by point rebuttal (*Dodel et al.*)

Comments to all reviewers:

We thank all the reviewers for the time and effort they put in reviewing our manuscript and providing us with very useful and constructive comments. We believe we have now addressed all their concerns as detailed below in our point-by-point rebuttal, and hope that in light of these changes, they find our manuscript acceptable for publication in *Nature Methods*. Please note that for the ease of assessing our revisions, in addition to listing the changes in this point-by-point rebuttal, we have also highlighted any areas with significant changes to the text, in order to facilitate the review of our revised manuscript.

In addition to the changes resulting from the reviewers comments, we also implemented a number of other corrections and changes to the manuscript. These additional corrections are as follows:

1. A new author (Zeinab Rekad) is now added to the authors list, as she carried out some of the revision experiments.
2. The number of identified U1 subunits in one of the comparison studies (*Chu et al., 2012*) listed in Fig. 1F was actually higher than what we had initially reported (4 out of 10, instead of 1 out of 10). This mistake arose due to a discrepancy in the reported protein nomenclatures, but has now been corrected.
3. In Figure 2A, the number of ND4 targeting oligos was wrongly stated as 8, but this should have been 9. This has now been corrected.
4. PRIDE accession for the *NORAD* ND4 TREX experiment was not initially included in the data availability section by mistake. This has now been rectified (see data availability).
5. Manuscript line 489: (381 out of 400) was reported incorrectly. The correct count should have been (378 out of 396). This has now been fixed.
6. For the sake of consistency across figures, all experimental validation data have now been moved to the supplementary figures, with TREX data forming the bulk of main figures. As a result, Fig 6E from the original manuscript has now moved to S6C.
7. Additional sections for RNA Affinity Estimations, CLIP, and Statistical Analysis have now been added to the Materials and Methods.

Point-by-point response to individual reviewer comments:

Reviewer #1:

Remarks to the Author:

The authors describe, TREX, a novel approach for identifying the proteins binding an endogenous RNA of interest within cells. The approach builds on previous methods to isolate crosslinked protein-RNA aggregates and extends it by utilizing RNase-H-mediated release of the proteins associated with a specific region. This is a clever idea and it appears to work very well, at least on relatively abundant RNAs. The approach addresses a very important need in the community for methods that can identify proteins binding to a specific region within a long RNA. This is an important problem as many RBPs are promiscuous and so the set of proteins that bind the entire RNA is often very large and it is very difficult to distill from it the proteins that are functionally relevant for further interrogation. The method thus addresses an urgent and unmet need in the community, and I expect it to become widely used, and so it is a great fit for Nature Methods.

> We greatly appreciate the reviewer highlighting the important unmet need in the RNA community that our method addresses.

The paper is well written, and the presented results appear to be convincing and give some interesting insights into the biology of both NORAD and rRNAs, giving leads to further studies. Overall, I found the level of detail describing these results to be too high, and the discussion is quite repetitive, mainly going over the many proteins found in each part and referencing other studies, so the manuscript can be shortened quite substantially giving more focus to the new method, and to the particular new insights obtained, and making the text more readable (potentially big chunks of the text can become Supplementary Notes).

> We have now taken this suggestion on board and have tried to shorten the main text where the individual names and details of some of the interactors were being discussed.

I have few specific requests:

- The authors should indicate in the Results section the number of cells used in each TREX experiment (including for NORAD, rRNA etc.). Where applicable, it can also be compared with the number of cells used in previous RAP-MS studies (e.g., for NORAD).

> The number of cells used for each experiment is now clearly stated in the text of the revised manuscript (see manuscript lines 137, 192, 272, 311, 321, 364, & 387), as well as in the material and methods section (see manuscript lines 662-664). With regards to the comparison with previous RAP-MS studies, the only comparable target is *UI*, for which the comparison is shown in Figure 1F. Although we have used less cells in our *NORAD* TREX experiments, compared to the previous pulldown-based studies, these experiments are not directly comparable as our experiments only target a section of *NORAD*, rather than the full RNA. Therefore, a head-to-head comparison of cell numbers in these experiments may not be strictly accurate, so we have refrained from stating this in the manuscript.

- The authors discuss the possible relationship between TOP1 they identify as binding ND4 in *NORAD* and RBMX which was reported previously by Munschauer et al., but it is important to mention that RBMX binding site is quite far from ND4, and is rather at the 5' end of *NORAD*.

> We thank the reviewer for raising this important point. Indeed, RBMX was previously reported to primarily bind to the first 900 nucleotides of the *NORAD*¹, although looking at this published CLIP-seq data, one can also see a small but distinct binding peak within the ND4 region (see Rebuttal Figure 1). In line with these results, our new CLIP-qPCR analysis demonstrates that similar to PUM1, RBMX can make contacts with the ND4 region (see revised Fig S2F-H, and manuscript lines 222 to 225). As mentioned in our response to reviewer 3, we also carried out CLIP-qPCR validation for the TOP1, confirming its direct binding to the ND4 region (see revised Fig. S2I-J). It is worth noting, however, that the levels of *NORAD* ND4 enrichment in PUM1 immunoprecipitates are significantly higher than those of RBMX and TOP1, potentially indicative of the superior binding affinity of PUM1 to this RNA segment (see response to reviewer 3 comments on the discussion of affinity).

Collectively, these CLIP-qPCR analyses add further validation to our TREX results.

Rebuttal Figure 1: CLIP-seq crosslink tracks corresponding to *NORAD* from RBMX, FUBP1, FUBP3, and PUM1 immunoprecipitates, taken from Munschauer et al 2018¹. The ND4 region is marked by the red rectangle.

- I found the annotation enrichment presentation in Fig. 4C,F, Fig. 5C-I and elsewhere to be quite confusing, and the relative position of the category on the X/Y axis doesn't really mean much. A simpler presentation of the most enriched categories, color-coded by the significance would make things simpler yet easier to comprehend.

> The choice of this type of category enrichment presentation is based on being able to visualise not only the enrichment levels and their statistical significance (i.e. X and Y axes), but also the category sizes (depicted by the size of the circles). This is an often ignored factor in enrichment analyses, yet quite insightful to know; a small category of proteins being highly enriched in a selection may not be as biologically impactful as a very large category of proteins being enriched to the same level and significance. It is for this reason that we have opted to show our category enrichments in this way. However, we appreciate that some of these graphs are just too busy and therefore somewhat confusing for our readers. Therefore, in the revised manuscript we have tried to limit displaying the names for categories that are too similar, as this does not add much further value, and make these graphs less busy looking (see revised Fig. 2E, 4F, and 5C). We hope that this change makes the graphs less confusing for readers.

- Line 467 - “more than 95% of the interacting proteins (381 out of 400) displayed highly specific positional interaction patterns” – how is the specificity defined here? What does it mean that the binding was “highly specific”? What are the effect sizes and the significance of the specificity?

> We appreciate that the writing here was not very clear, so we have now rewritten this part (see manuscript lines 488-490). We simply meant that these proteins seem to be primarily bound to one section of 45S rRNA (see revised Fig. 6A).

- Sup figS1 D-G: please add normalization to other parts of the same transcript to show the specificity of degradation by probes.

> This is now done in the revised manuscript (see revised Fig. S1D-F and legends).

- Was total lysate (line 690) prepared from crosslinked or non crosslinked sample, please clarify in text.

> Apologies for not making this clear. The total lysate was indeed also prepared from crosslinked cells. This is now stated clearly in the revised manuscript (see manuscript line 704).

Reviewer #2:

Remarks to the Author:

This manuscript describes a novel method to characterize the proteins bound to specific regions of a target RNA. It does this by first isolating RNA-RBP complexes from crosslinked cells and then releasing the proteins bound to target regions by incubating with antisense DNA and digesting RNA-DNA hybrids using RNaseH. The robustness of this method is established through experiments profiling snRNAs, lncRNAs, and rRNAs. The analysis of different regions of 45S-rRNA and spacers regions of 45S pre-rRNA highlights the power of this technology and how it breaks through current limitations in identifying RNA-RBP interactions. Overall, the paper contains high-quality data, the results are significant, and the claims are generally supported. However, a few issues need to be addressed before this paper can be accepted.

1. In the authors' description of TREX in lines 91-93, it is not clear that they isolate the RNA-protein adduct interface and resolubilize before adding the tiling DNA antisense molecules. This makes the description of the method a bit confusing. The authors should consider amending this to ensure it is clear to the reader.

> We thank the reviewer for highlighting this point. We have now amended this part of the text accordingly to make the procedure more clear (see manuscript lines 91-94).

2. When introducing U2 snRNA as well as the ribosomal RNA molecules and spacer regions profiled, the authors may want to consider discussing the length of the RNAs and the regions they profile. The authors should consider adding this as it would give a better sense of the capabilities of the technology.

> Great suggestion. This is now added for every TREX experiment shown (see manuscript lines 133, 181, 260, 307, & 383).

3. Similarly, the authors should discuss the length of the DNA tiles they selected and justify why they selected this length. While this is accessible in the supplement, considering this is a paper focused on showcasing a new method, this would be important to elaborate on.

> The average length of DNA probes used in TREX (~60 nucleotides) was modelled based on the average probe sizes used in RNase H-mediated ribodepletion methods for RNA-sequencing library preparations^{2, 3}. This information is now added to and briefly discussed in the main text of the revised manuscript (see manuscript lines 95-98).

4. One of the major advantages the authors claim that TREX has over RNA-centric methods profiling RNA-RBP interactions is input amount. The authors specifically claim that TREX was able to achieve superior performance with as low as 5 to 10% of the input required by other methods. I believe this claim is overstated. The authors used their ability to profile U1 snRNA from 10 million cells to support their assertion that they have greater sensitivity than others. They cited three other papers that also profiled U1 but that employed more than 100 million cells for that purpose. However, a huge factor in determining the number of cells that are required for successful analysis of a particular RNA target is its cellular abundance. All three of the papers cited characterizing the U1 snRNA interactome used it as a control experiment because its interactome is well-characterized and the U1 snRNA is highly abundant (> 1 million copies per cell). All three papers then used the same number of cells (>100 million) to target lncRNAs, which are generally much less abundant.

When the authors of this paper went after a lower abundance lncRNA target (NORAD, ~1000 copies per cell), they also required 100 million cells, similar to the three other studies. They were successful using 10 million cells for U1 snRNA because of its high abundance, but similarly to the

other studies, needed 100 million cells for low abundance targets. Thus their claim of greater sensitivity is not well-supported.

> We thank the reviewer for raising this point. We agree with them that it is better not to make the claim of greater sensitivity, as the purpose of the previous studies mentioned was indeed not testing the lower limits of pulldown based assays. Therefore, we have now taken this claim out from our results and discussion. We do, however, believe it is still valid to keep Figure 1D, as a demonstration that more of the known components of U1 complex were detectable in our TREX analysis, even though this was from less input material. In other words, although the previous pulldown based studies could have potentially gone lower in terms of their cell numbers (so sensitivity is not strictly comparable between the studies), TREX seems to have a higher recall, even from less input material.

In general, this paper is excellent and the technology to assess sequence specific RNA-RBP interactions is interesting and important, and the authors make a rigorous case for their method. However, I strongly believe that the claims of greater sensitivity are overstated. I would recommend the authors remove this claim as it is not needed to make a case for their work and may in fact weaken the paper by reducing its credibility.

> We appreciate the reviewer highlighting the importance of our method, and recognising the rigor of our work. As mentioned above, we have now taken on board their suggestion and have removed the claims of greater sensitivity from our revised manuscript.

Reviewer #3:

Remarks to the Author:

In the manuscript "TREX reveals proteins that bind to specific RNA regions in living cells", Dodel, Giuducci, and colleagues introduce TREX, a method designed to isolate and identify RNA binding proteins interacting with specific regions of a given RNA molecule of interest. The described method is an extension of a set of related protocols that enrich for crosslinked and denatured ribonucleoprotein complexes at an organic/aqueous interface and allow for the global identification of RNA binding proteins (PTex/OOPS/XRNAX). The authors add a step to this protocol by recovering denatured RNPs from the interphase and specifically degrading defined stretches of RNA by hybridizing antisense DNA oligos and treating with RNase H. RBPs that were

interacting with the RNA stretch of interest are now free to migrate into an organic phase and can be identified by mass spectrometry. The authors benchmark this elegant approach by identifying RBPs interacting with U1 RNA, a ~500 nt stretch of the well-characterized NORAD lncRNA, as well as the entire 45S rRNA (that contains the 5.8S, 18S, and 28S rRNAs, as well as linker regions). Overall the method appears promising in that it may indeed enrich for RBPs that were interacting with the regions of interest, as they do recover previously known interactors. The approach is straightforward and appears to be reproducible and adaptable for use in most molecular biology laboratories (that have access to mass spectrometry, of course). That said, enthusiasm for the manuscript is dampened by the incomplete characterization of specificity and recovery of noise throughout the manuscript. As their approach will most likely be used to try to identify specific regulators of interesting RNA molecules, careful and critical analysis of the false discovery rates is crucial to demonstrate usefulness.

1. the authors recover most of the known interactors of U1. Considering that U1 is a very abundant RNA, question is why are two of the canonical snRNPs missing? Can they be identified by mass spec in principle (trypsin digestion problems or the like)?

> Mass spectrometry based peptide identification and quantification is stochastic in nature, and depends on multiple factors such as protein abundance, the number of theoretically available unique tryptic peptides, and the ability of the generated peptides to fly in the gas phase. All of these factors play a crucial role on whether a protein can be confidently identified and quantified in proteomics samples (more high score peptides = the better protein quantification).

The specific *U1* subunits not detected in our analysis were SNRPD1 and SNRPF. Interestingly, the latter was also never identified as a *U1* binder in any of the previous pulldown based *U1*-interactome studies, while the former was only identified in 1 out of the 3 studies (See Rebuttal Fig. 2). This observation strongly indicates that SNRPD1 and SNRPF are for whatever reason hard to identify as specific *U1* interactors, irrespective of whether TREX or pulldown based assays are used.

—————	Chu et al	McHugh et al	Desideri et al.	TREX
SNRPA	✓	✓	✓	✓
SNRPB		✓		✓

SNRPC		✓	✓	✓
SNRPD1	✓			
SNRPD2	✓	✓	✓	✓
SNRPD3	✓	✓	✓	✓
SNRPE		✓		✓
SNRPF				
SNRPG			✓	✓
SNRNP70		✓	✓	✓

Rebuttal Figure 2: List of U1 spliceosome subunit proteins and whether they were identified (✓) as specific *UI* interactors in TREX vs. three previous *UI* RNA affinity capture-MS studies mentioned in manuscript Fig. 1F.

The exact reasons for this subunit-specific behaviour are not fully clear to us and probably constitute a combination of points raised above. Ultimately, the number of confidently identified peptides is likely to be a significant factor, since both of these subunits were identified with the least number of peptides in our TREX experiments, compared with the other subunits (Rebuttal Fig. 3). This means that the quantification of – vs + RNase H conditions is likely not as accurate for these two subunits compared to the others, thus providing a potential explanation for their lack of significant enrichment as specific *UI* binders.

	Number of identified peptides in total
SNRPA	9
SNRPB	7
SNRPC	5
SNRPD1	3
SNRPD2	7
SNRPD3	6
SNRPE	4

SNRPF	3
SNRPG	5
SNRNP70	33

Rebuttal Figure 3: The number of identified for each U1 spliceosome subunit protein in the *U1* TREX experiment (Dataset S1). SNRPD1 and SNRPF, which failed to come up as significantly enriched in the *U1* digested samples, both happen to have been detected by 3 peptides only, which is lower than all the other subunits.

Nevertheless, we feel it is important to highlight that TREX is still outperforming pulldown based methods in terms of identifying known *U1* binders. In fact, in our TREX analysis, we identify 8 out of 10 U1 spliceosome subunits, which is more than all the previous pull-down based studies (Fig. 1D & Rebuttal Fig. 2). Therefore, in terms of recall rate, TREX supersedes the existing state-of-art.

The authors do not comment on the proteins that are not snRNPs and are not part of the splicing machinery - why are they enriched with the U1 RNP? Is this a universal background that they also find in the other datasets?

> There are indeed a number of novel interactors that we have identified for *U1*, but mostly at lower enrichment levels than the known *U1* spliceosome subunits (see Fig. 1C and Dataset S1). However, these do not appear to be background or non-specific binders that happen to be also found in other TREX datasets. In fact, the overlap between significant TREX-revealed interactors of *U1* (27 proteins – Dataset S1) and *NORAD* ND4 region (360 proteins – Dataset S2) is only two proteins (ADAR and SNRNP70, both of which are validated interactors of *U1* and not the additional proteins the reviewer is enquiring about – see Fig. 1D). Moreover, the overlap between the *U1* and the full-length *45S* analysis (160 proteins – Dataset S3), is zero.

Remarkably, even when interactors of different regions of *45S* rRNA were compared against each other, only 18 out of 396 proteins seem to be consistently enriched in all sections (Fig. 6A). These 18 proteins are also not some non-specific background. In fact, we go to some length to show that their binding to the full-length of *45S* rRNA is real and likely a novel functional feature of these proteins in rRNA metabolism (Fig. 6 & S6).

Together, these lines of evidence strongly suggest that the additional novel proteins that we report in our different TREX analyses are not some universal background, but rather very specific sets of

proteins that are uniquely released from each targeted RNA region. In this respect, TREX seems to be more specific compared to RAP-MS, which has been reported to suffer from a recurring set of >200 non-specific universal background proteins^{1, 4-6}.

2. the authors analyze a fairly long (~500 nt region) of NORAD and recover hundreds of proteins as potential interactors. Among them are the known interactors (PUM1/2) that the Mendell lab found as well as RBMX/SAM68/IGF2BP - interactors that the Lander found in a conflicting manuscript. It appears clear from the reading of the literature that the interaction with PUM proteins is best characterized, as there were plenty of follow-up studies - also consistent with the mainly cytoplasmic localization of the lncRNA. Why do the authors recover what are mainly nuclear RBPs (e.g. RBMX, and plenty of others in the dataset).

> We would like to clarify that our detection is not limited to mainly nuclear RBPs. In contrast, our study also captures a substantial number of cytoplasmic RBPs, including but not limited to IGF2BP2, IGF2BP3, SERBP1, YBX3, DHX30, CNBP, TIAL1, and PUM1, as detailed in Dataset S2. Complementing this, our analysis of the Gene Ontology Cellular Compartment (GOCC) annotations indicates that the interactors of ND4 are evenly split between nuclear and cytoplasmic compartments, as depicted in Figure S2D. We acknowledge the ongoing controversies about the mechanisms of action of *NORAD*. However, regarding its interactome, our findings are consistent with the majority of the published studies showing that both cytoplasmic as well as nuclear proteins making contact with this lncRNA^{1, 7-9}.

It is pertinent to note that our investigations in this manuscript solely centre on the interactome and does not extend to the functional assessment of *NORAD* mechanism. As such, determining the functional significance of the interactions identified—both known and novel—within the context of *NORAD* biology falls beyond the scope of our current work, which is strictly focused on methodology. On this point, please also see text changes made to the *NORAD* section of our manuscript, in which we have now made sure that our discussions do not stray into speculations on function (manuscript lines 170-255).

Perhaps performing TREX from nuclear and cytoplasmic extracts could clarify whether these interactions just reflect different compartments that *NORAD* could localize in (and thereby also clarify some contention in the literature).

> Although this experiment is very interesting from the perspective of *NORAD* biology, in our opinion, the question of delving more into the detailed workings of *NORAD* and clarifying some

of the contentions in the literature goes beyond the scope of our manuscript. In the revised manuscript, we have now gone to some length trying to assess some of the complexities of the *NORAD* interactome (see for instance the response to the next point), but engaging significantly more with these complexities and the associated debates in the field would, we believe, detract from the focus of our article, which is methodology. We are, however, certain that our findings can be utilised by researchers in the *NORAD* field to get at some of these questions and complexities, if they wish to do so.

Very concerning is that the volcano plot shows moderate enrichment for PUM1 - why are other proteins so much more enriched? Is that reflective of real binding? Some validation should be performed to convince the reader that we are not dealing with background.

> We would like to clarify here that the fold enrichments reported in our TREX analyses should not be misconstrued as a measure of binding superiority of a protein to a specific RNA region. The enrichment levels we report provide a quantitative snapshot of the prevalence of proteins on a particular RNA region, reflecting an RNA-centric perspective of the interactome. This is not to be interpreted as a measure of protein binding affinity or efficiency. In fact, enrichment is influenced by other factors beyond mere affinity, such as the overall abundance of a protein. Our observations indeed suggest that protein abundance plays a significant role in the degree of enrichment values observed. This is exemplified when correlating the enrichment values of the hits from our ND4 TREX analysis (extracted from Fig. 2B), with their respective iBAQ-calculated absolute protein abundance values (extracted from Fig. 2C), which shows a significant correlation (Rebuttal Fig. 4A). This underscores the influence of protein abundance on our RNA-centric enrichment metrics.

A noteworthy observation from this analysis is the behaviour of PUM1. Despite its relatively lower abundance in our cells in comparison to the other hits, PUM1 exhibits a decent level of enrichment in the ND4 TREX, suggesting it is likely a potent interactor. Conversely, despite RBMX being more abundant than PUM1, its enrichment is approximately ~2-fold lower in the TREX, suggesting a potential lower affinity for the ND4 region (Rebuttal Fig. 4A).

After observing this, we wondered whether we could utilise iBAQ-based absolute abundance measurements to estimate the relative affinity of the significant hits in a TREX experiment for their target RNA. Since RNase-H mediated depletion in TREX is almost complete (i.e. stoichiometric), this should be possible through calculating the ratio of absolute abundance in TREX, relative to the matching whole cell extracts:

$$\text{Affinity of a protein for RNA } (K_a) = \frac{[RNA \text{ bound}]}{[Total]} = \frac{iBAQ \text{ TREX}}{iBAQ \text{ total}}$$

(Where the *iBAQ TREX* is calculated from estimated iBAQ values in the RNase-H digested TREX runs, subtracted by the negative control TREX runs).

When we calculate this value for each of the significant ND4 interacting proteins, we indeed observe that PUM1 is amongst one of the highest affinity binders of ND4 (Rebuttal Fig. 4B). We believe this is strongly in line with (and important for) the proposed ability of *NORAD* to sequester PUM1, particularly when considering that the ND4 region only contains 4 Pumilio response elements (PREs), with a further 14 PREs present throughout the rest of the transcript. We now present and discuss these new results in the revised manuscript (see revised Fig. 2D and manuscript lines 199-205).

We must again stress that all these observations solely centre on the interactome of *NORAD* and do not extend to the functional assessment of *NORAD* mechanism, and any investigation of the mechanistic significance of these results falls beyond the scope of our current work. Nevertheless, we believe these results demonstrate another novel and unique ability of TREX to provide an estimate of binding affinity, when combined with matching total lysate analysis, thanks to the stoichiometric extraction of RNA-bound proteins in TREX. It is for this reason that we believe it is useful to add them to the revised manuscript.

Rebuttal Figure 4: (A) Correlation between the *NORAD* ND4 TREX enrichment levels and iBAQ-estimated total protein abundance values in HCT116 cells, for the identified significant interactors. Known *NORAD* interactors are highlighted in purple. Pearson correlation coefficient (CC) and the significance p-value of the correlation are both reported on the graph. (B) The relative affinities of the identified interactors for ND4 RNA were calculated using background normalised iBAQ-estimated protein abundance values from the ND4 TREX experiment, divided by the iBAQ estimated total protein abundance from an equivalent amount of HCT116 whole cell lysate, and plotted as a ranked plot. PUM1 and IGF2BP3 are amongst the top high affinity binders of the ND4 region.

3. In all their experiments, the authors also identify previously unappreciated interactors for the RNAs of interest - even rivalling or exceeding enrichment of known interactors. The authors should validate these interactions - not with RIP, as this recovers secondary interactions and does not isolate the specific binding sites - but with CLIP. CLIP as an orthogonal approach can demonstrate that an identified RBP indeed bound at a sequence element in the RNA region digested by RNase H.

> We appreciate this great suggestion and have now incorporated such CLIP-based validations into the revised manuscript. We strategically selected two interactors identified in two of our TREX analyses for further investigation through CLIP-PCR.

The first candidate we chose was Topoisomerase-I (TOP1), due to its notable enrichment in our *NORAD* analysis (see Fig. 2F), as well as the background literature. In brief, although TOP1 has been recently implicated in *NORAD* function (the article mentioned earlier by the reviewer), a direct interaction between the two was not been demonstrated¹. Instead, it has been proposed that *NORAD* interacts with TOP1 indirectly via RBMX. Contrary to this proposal, our TREX data

suggests a direct interaction between TOP1 and *NORAD* — a finding that we have now further substantiated through our new CLIP-qPCR validations (see revised Fig. S2I-J and manuscript lines 222-233). As mentioned in our response to reviewer 1, for comparative purposes, we also confirmed the binding of RBMX and PUM1 via CLIP (revised Fig. S2F-H), which further underscore the high affinity nature binding of PUM1 to *NORAD*.

The second interactor we selected for validation was UBR7, following its identification as a highly prominent interactor of the 5.8S rRNA (Fig. 4D). Given the concise and well-characterised interactome of 5.8S rRNA, UBR7 was an attractive candidate for further validation. Our subsequent CLIP-qPCR analysis confirmed UBR7's interaction with premature 5.8S rRNA (see revised Fig. S4F-G and manuscript lines 328-333). Moreover, we explored the functional significance of UBR7 by assessing the effects of its depletion on ribosome biogenesis, uncovering a specific involvement in 5.8S rRNA maturation (see revised Fig. S4H-I & manuscript lines 333-338). These results propose UBR7 as a novel 5.8S biogenesis factor.

We would also like to highlight that in addition to these CLIP-qPCR validations, we provide validation in form of re-analysis of existing eCLIP data for NOLC1, GTF2F1, and SRSF9 (Fig. S6A). These proteins were identified as binders to multiple regions of the 45S rRNA (Fig. 6A). By re-analysing ENCODE consortium eCLIP data^{10, 11}—this time mapped against the human rDNA sequences that are traditionally excluded from standard CLIP-seq analyses—we could clearly demonstrate a binding pattern across the entire length of 45S rRNA, which is in full agreement with our TREX findings.

Collectively, we believe that the above mentioned eCLIP and CLIP-qPCR analyses provide significant orthogonal validation of some of our most novel and interesting TREX findings, thus giving further confidence to the method.

4. to be very useful in the identification of regulatory RBPs - rather than recovering a substantial amount of RBPs, as in the case of *NORAD* - the authors could use shorter tiles; e.g. in *NORAD* target one of the PREs and show recovery of PUM1 and a region with no PRE and show that no Pumilio proteins are identified by TREX.

> We thank the reviewer for raising this point. However, we must highlight the inherent specificity challenges when aiming to target short sequences (e.g. an 8-nucleotide motif corresponding to a single PRE). Antisense oligos of such size will inevitably be non-specific due to the high probability of their complementary sequences recurring throughout the transcriptome. The only

way to overcome this issue would be to target a wider segment (thus ensuring specificity), but with the short sequence of interest, included or not, at the end the targeted segment. This could in theory work for targeting a single PRE in *NORAD*, but we would like to point out that we have already demonstrated such capability for TREX in our 5'ETS and 3'ETS analyses (see Fig. S5E & Fig. S5I), where two known end-region binders (*RSP9* and *RPL17*) were specifically revealed in our ETS analyses, even though the sequences covering their binding sites were only a few nucleotides-long at the end of the targeting tiled sections.

Rebuttal references

1. Munschauer, M. *et al.* The *NORAD* lncRNA assembles a topoisomerase complex critical for genome stability. *Nature* **561**, 132-136 (2018).
2. Archer, S.K., Shirokikh, N.E. & Preiss, T. Selective and flexible depletion of problematic sequences from RNA-seq libraries at the cDNA stage. *BMC genomics* **15**, 401 (2014).
3. Morlan, J.D., Qu, K. & Sinicropi, D.V. Selective depletion of rRNA enables whole transcriptome profiling of archival fixed tissue. *PLoS One* **7**, e42882 (2012).
4. McHugh, C.A. *et al.* The *Xist* lncRNA interacts directly with SHARP to silence transcription through HDAC3. *Nature* **521**, 232-236 (2015).
5. Chu, C. *et al.* Systematic discovery of *Xist* RNA binding proteins. *Cell* **161**, 404-416 (2015).
6. Desideri, F. *et al.* Intronic Determinants Coordinate *Charme* lncRNA Nuclear Activity through the Interaction with MATR3 and PTBP1. *Cell reports* **33**, 108548 (2020).
7. Lee, S. *et al.* Noncoding RNA *NORAD* Regulates Genomic Stability by Sequestering PUMILIO Proteins. *Cell* **164**, 69-80 (2016).
8. Tichon, A. *et al.* A conserved abundant cytoplasmic long noncoding RNA modulates repression by Pumilio proteins in human cells. *Nat Commun* **7**, 12209 (2016).
9. Spiniello, M. *et al.* HyPR-MS for Multiplexed Discovery of MALAT1, NEAT1, and *NORAD* lncRNA Protein Interactomes. *J Proteome Res* **17**, 3022-3038 (2018).
10. Consortium, E.P. An integrated encyclopedia of DNA elements in the human genome. *Nature* **489**, 57-74 (2012).
11. Luo, Y. *et al.* New developments on the Encyclopedia of DNA Elements (ENCODE) data portal. *Nucleic acids research* **48**, D882-D889 (2020).

Decision Letter, first revision:

Our ref: NMETH-A53590A

29th Nov 2023

Dear Dr. Mardakheh,

Thank you for submitting your revised manuscript "TRES reveals proteins that bind to specific RNA regions in living cells" (NMETH-A53590A). It has now been seen by the original referees and their comments are below. The reviewers find that the paper has improved in revision, and therefore we'll be happy in principle to publish it in Nature Methods, pending minor revisions to satisfy the referees' final requests (see Reviewer3's comments) and to comply with our editorial and formatting guidelines. In the mean time, it would be great if you can send me a rebuttal letter to address these final requests.

TRANSPARENT PEER REVIEW

ORCID

Sincerely,
Lei

Lei Tang, Ph.D.
Senior Editor
Nature Methods

Reviewer #1 (Remarks to the Author):

The authors have addressed my comments from the previous round of review in a satisfactory way, and I believe the manuscript has improved. I can now recommend publication in Nature Methods.

Reviewer #2 (Remarks to the Author):

This paper describes a novel method which uses antisense DNA to capture specific RNAs and then employs RNaseH mediated degradation to specifically look at proteins bound to the specific regions targeted by the antisense DNA. The authors profile a variety of RNAs including snRNAs, lncRNAs, and rRNAs. Comparative analysis of different regions of 45S-rRNA and spacers regions of 45S pre-rRNA highlight the power of this technology. The novel technology is a significant breakthrough in RNA-protein interactomics and is robustly validated. In light of the changes that the authors made to the manuscript and the additional experiments performed, I believe this paper should be accepted for publication in Nature Methods.

Reviewer #3 (Remarks to the Author):

The authors did a great job with the changes they made to the manuscript as well as with discussing the reviewers' concerns. I think that the method will be a good addition to the arsenal of interactome-capture type mass spec-based approaches. That said, I disagree with the authors' assertion that following up on the data regarding the NORAD interactome is beyond the scope of the manuscript - the authors want to convince the reader that the data derived from the method can be useful to provide insight into biological questions - why else would one want to use it? Therefore it would be good to show in a complex system (such as NORAD) that the data are indeed useful and allow for prioritization of interesting hits to follow up; hence the disappointment that the best characterized interactor of NORAD, Pum1 was not a top hit, and hence the concern about the many nuclear proteins in the interactome of a predominantly cytoplasmic RNA. Also, when I suggested looking into smaller footprints, particularly in NORAD, I did not mean 8-mers of a single PRE, but a smaller region with defined binders, e.g. in the Pub-binding region, that should not have too many other RBPs binding.

Author Rebuttal, first revision:

Reviewer #3 (Remarks to the Author):

The authors did a great job with the changes they made to the manuscript as well as with discussing the reviewers' concerns. I think that the method will be a good addition to the arsenal of interactome-capture type mass spec-based approaches.

> We are grateful for the reviewer's positive feedback and constructive criticism, which have been instrumental in enhancing our manuscript. We are very encouraged by their endorsement of our method as a valuable addition to interactome-capture mass spectrometry approaches.

That said, I disagree with the authors' assertion that following up on the data regarding the NORAD interactome is beyond the scope of the manuscript - the authors want to convince the reader that the data derived from the method can be useful to provide insight into biological questions - why else would one want to use it? Therefore it would be good to show in a complex system (such as NORAD) that the data are indeed useful and allow for prioritization of interesting hits to follow up; hence the disappointment that the best characterized interactor of NORAD, Pum1 was not a top hit, and hence the concern about the many nuclear proteins in the interactome of a predominantly cytoplasmic RNA.

> We would like to clarify that our intention was not to evade relevant discussions regarding *NORAD*'s function, but to maintain a degree of focus that is appropriate for a methods paper. Nonetheless, we agree that our interactome data offers significant insights into *NORAD* biology and function, for those who are interested in this lncRNA. For instance, our data on direct interaction of *NORAD* with TOP1, rather the previously suggested model of indirect recruitment via RBMX¹, could be potentially relevant in addressing some of the contentions in the field², considering the known function of TOP1 in regulation of genomic stability³. Moreover, the new data added to the revised manuscript regarding the estimation of protein affinities, made possible by the unique ability of TREX to achieve near-complete depletion of its targeted RNA regions, is likely to provide important insight into how functionally important binders can be further distinguished, depending on the specific biological query: if one is interested in understanding how a protein is regulated by an RNA (i.e. ribo-regulation), it would be important to know what fraction of the protein in question is bound to the RNA, as opposed to just determining whether a significant degree of binding occurs between the two (as showcased for PUM1-*NORAD* interaction in our study).

We have further modified the text of our discussion (Revised manuscript lines 431 - 435) to emphasise this important point more clearly (which is highly relevant for hits prioritisation as the reviewer highlights).

Also, when I suggested looking into smaller footprints, particularly in NORAD, I did not mean 8-mers of a single PRE, but a smaller region with defined binders, e.g. in the Pub-binding region, that should not have too many other RBPs binding.

> We thank the reviewer for elaborating on their suggestion regarding smaller footprint analysis. As highlighted in our initial response, TREX's application for analysing specific smaller segments with well-defined binding partners was showcased in our 5'ETS and 3'ETS analyses. So while not demonstrated explicitly for NORAD, our study effectively shows this capability of TREX, further highlighting the versatility of our methodology in addressing a range of detailed questions regarding RNA-protein interactome analysis.

References:

1. Munschauer, M. *et al.* The NORAD lncRNA assembles a topoisomerase complex critical for genome stability. *Nature* **561**, 132-136 (2018).
2. Elguindy, M.M. *et al.* PUMILIO, but not RBMX, binding is required for regulation of genomic stability by noncoding RNA NORAD. *Elife* **8** (2019).
3. Tuduri, S. *et al.* Topoisomerase I suppresses genomic instability by preventing interference between replication and transcription. *Nature cell biology* **11**, 1315-1324 (2009).

Final Decision Letter:

Dear Dr Mardakheh,

I am pleased to inform you that your Article, "TREX reveals proteins that bind to specific RNA regions in living cells", has now been accepted for publication in Nature Methods. The received and accepted dates will be 21st Aug 2023 and 16th Jan 2024. This note is intended to let you know what to expect from us over the next month or so, and to let you know where to address any further questions.

Over the next few weeks, your paper will be copyedited to ensure that it conforms to Nature Methods style. Once your paper is typeset, you will receive an email with a link to choose the appropriate publishing options for your paper and our Author Services team will be in touch regarding any additional information that may be required. It is extremely important that you let us know now whether you will be difficult to contact over the next month. If this is the case, we ask that you send us the contact information (email, phone and fax) of someone who will be able to check the proofs and deal with any last-minute problems.

After the grant of rights is completed, you will receive a link to your electronic proof via email with a

request to make any corrections within 48 hours. If, when you receive your proof, you cannot meet this deadline, please inform us at rjsproduction@springernature.com immediately.

Please note that *Nature Methods* is a Transformative Journal (TJ). Authors may publish their research with us through the traditional subscription access route or make their paper immediately open access through payment of an article-processing charge (APC). Authors will not be required to make a final decision about access to their article until it has been accepted. [Find out more about Transformative Journals](https://www.springernature.com/gp/open-research/transformative-journals)

Please note that you and any of your coauthors will be able to order reprints and single copies of the

issue containing your article through Nature Portfolio's reprint website, which is located at <http://www.nature.com/reprints/author-reprints.html>. If there are any questions about reprints please send an email to author-reprints@nature.com and someone will assist you.

Best regards,
Lei

Lei Tang, Ph.D.
Senior Editor
Nature Methods